

# A geostatistical spatially varying coefficient model for mean annual runoff that incorporates process-based simulations and short records

Thea Roksvåg[1, 2], Ingelin Steinsland[1], and Kolbjørn Engeland[3]

[1]Norwegian University of Science and Technology, NTNU, Høgskoleringen 1, 7491 Trondheim, Norway
[2]Norwegian Computing Center, NR, Gaustadalléen 23A, 0373 Oslo, Norway
[3] The Norwegian Water Resources and Energy Directorate, NVE, Middelthuns gate 29, 0368 Oslo, Norway

**Correspondence:** Thea Roksvåg (roksvag@nr.no)

**Abstract.** We present a Bayesian geostatistical model for mean annual runoff that incorporates simulations from a process-based hydrological model by treating the simulations as a covariate in the statistical model. The regression coefficient of the covariate is modeled as a spatial field such that the relationship between the covariate (simulations from a hydrological model) and the response variable (observed mean annual runoff) is allowed to vary within the study area. Hence, it is a spatially varying

coefficient. A preprocessing step for including short records in the modeling is also suggested and we obtain a model that can exploit several data sources by using state of the art statistical methods.

The geostatistical model is evaluated by predicting mean annual runoff for 1981-2010 for 127 catchments in Norway based on observations from 411 catchments. Simulations from the process-based HBV model on a 1 km × 1 km grid are used as input. We found that on average the proposed approach outperformed a purely process-based approach (HBV) when predicting runoff

for ungauged and partially gauged catchments: The reduction in RMSE compared to the HBV model was 20 % for ungauged catchments and 58 % for partially gauged catchments, where the latter is due to the preprocessing step. For ungauged catchments the proposed framework also outperformed a purely geostatistical method with a 10 % reduction in RMSE compared to the geostatistical method. For partially gauged catchments however, purely geostatistical methods performed equally well or slightly better than the proposed combination approach. It is not surprising that purely geostatistical methods perform well

in areas where we have data. In general, we expect the proposed approach to outperform geostatistics in areas where the data availability is low to moderate.

## 1   Introduction

Runoff is defined as the flow of water that is generated from excess rainwater or meltwater, and that flows on the ground surface or within the soil towards a stream (WMO, 1992). Runoff indices of different types (annual runoff, seasonal runoff, maximum

runoff) are needed for a variety of purposes, e.g. for designing infrastructure, water supply and hydropower reservoirs, for assessment of water quality and ecosystems and for allocation of water resources between stakeholders. The temporal variability of runoff can also be used to study runoff's sensitivity to climate change. In spite of the large interest in runoff estimates,





the majority of the catchments in the world are ungauged, i.e. runoff measurements for deriving the relevant indices are not available and must therefore be predicted. This is known as the prediction of runoff in ungauged basins problem (PUB) and is

a key challenge in hydrology (Blöschl et al., 2013).

When predicting runoff in ungauged basins there are two main approaches: process-based approaches and statistical approaches. When taking a statistical approach, data from gauged catchments are used to develop a statistical relationship between the observed runoff and relevant variables like precipitation, temperature, land use and elevation. Next, the statistical relationship is used to make predictions for ungauged sites. Data-driven, statistical methods have been successfully used to

predict several flow indices in the literature (see e.g. Viglione et al. (2013); Merz and Blöschl (2005); Blöschl et al. (2013); Laaha and Blöschl (2005)), and in this article we consider a particular type of statistical models, namely geostatistical models. In geostatistical models it is assumed that locations that are close in space have more in common than locations that are located far away from each other, and this is formulated mathematically through a covariance function (see e.g Gelfand et al. (2010); Cressie (1993)). In the field of hydrology, the geostatistical Top-Kriging method proposed by Skøien et al. (2006) has been

shown to be particularly suitable for modeling catchment (areal) referenced data (Viglione et al., 2013), but other geostatistical approaches have also been suggested (Roksvåg et al., 2020; Sauquet et al., 2000).

Process-based hydrological models on the other hand, use physical relationships for e.g. conservation of mass and energy to simulate continuous flow series from which the flow index of interest can be derived. The input variables are as for the data-driven methods variables like precipitation, temperature and land use. Data from gauged catchments are used for validation

purposes and parameter calibration (see e.g. Doherty (2004); Lawrence et al. (2009)). The HBV model is an example of a process-based hydrological approach commonly used to estimate runoff in the Nordic counties (Bergström, 1976). Other process-based models are discussed in Blöschl et al. (2013); Clark et al. (2017); Fatichi et al. (2016).

In this article we suggest a geostatistical model for mean annual runoff that incorporates the simulations from a process-based model. The strength of process-based models is that they account for well-known, physical relationships between the input

variables (e.g. temperature and precipitation) and the output variables (e.g. runoff) and this way produce consistent hydrological estimates. The geostatistical approaches on the other hand provide uncertainty quantification and are typically better at ensuring a good fit between the runoff data and the model in areas where we have observations. However, the geostatistical estimates are often poor if the number of streamflow observations is low or if the underlying process is complex (Wang et al., 2017). Our working hypothesis is that a model that combines geostatistics with a process-based hydrological model will give better runoff

predictions than one of the model types alone.

There exist work based on similar ideas in the literature. One example is found in Pannecoucke et al. (2020) where the authors estimated the contamination level within the soil. This was done by using a process-based model to simulate flow a number of times and then computing empirical variograms based on the results. Next, the variograms were used for Kriging, which is a class of commonly used geostatistical models (see e.g. Cressie (1993); Gelfand et al. (2010)). In Laaha et al.

(2013) external drift Kriging was used for interpolation of streamflow temperatures where a physical relationship between mean annual stream temperature and stream gauge altitude was combined with the Top-Kriging approach. Considering models for mean annual runoff, we find a model in Qiu et al. (2018) where the authors combined a Budyko water balance model



with a geostatistical approach. In Sauquet (2006) mean annual runoff was estimated by a Kriging approach that is able to incorporate basin characteristics through a function $g(\cdot)$ and residual Kriging. The model was demonstrated with catchment

elevation as input, but the input to $g(\cdot)$ could also be simulations from a process-based model. The above papers all concluded that the geostatistical models that included process-based information gave better results than the alternative purely data-driven geostatistical approaches.

Following these, we suggest a Bayesian model for mean annual runoff where the observed runoff is used as the response variable and where mean annual simulations from a process-based hydrological model are incorporated through a covariate.

To connect the response variable (runoff) to the covariate (simulations from a process-based model), we use a *spatially varying coefficient* (SVC). Spatially varying coefficients have gained popularity in environmental modeling later years because there now exist computers and algorithms that are able to tackle the computational complexity they introduce (see e.g. Su et al. (2017); Finley (2011); Lu et al. (2009)). In a model with a spatially varying coefficient, the relationship between the response variable and the covariate is allowed to vary within the study area (Gelfand et al., 2003; Ferguson et al., 2009; Hastie and

Tibshirani, 1993), i.e. differently from a simple linear regression model where this relationship is restricted to be constant. The motivation behind using a spatially varying coefficient in our runoff model is that we assume that the process-based model is more accurate in some areas than others, and that the accuracy follows regional patterns.

There are several ways to implement a spatially varying coefficient. One possibility is to divide the study area into regions and let a given coefficient have one value for each region, like in e.g. Gamerman et al. (2003). However, this approach requires

that the user divides the study area into regions based on expert knowledge and this division is not always intuitive. An alternative approach where we avoid this issue, is to model the regression coefficient as a spatial field, more specifically a Gaussian random field (GRF) as described in e.g. Gelfand et al. (2003). Through the GRF, information about the regression coefficient at locations with data is regionalized to locations without data, following a spatial dependency structure. In this paper, we adopt the approach from Gelfand et al. (2003) and interpolate the relationship between the response variable (runoff

data) and the covariate (simulations from a process-based model) from gauged catchments to ungauged catchments. In addition to the spatially varying coefficient, we also include a standard spatial effect (GRF). This makes our model able to capture two different dependency structures, e.g. spatial dependency due to both short ranged and long ranged hydrological processes.

By our article, we aim to contribute towards finding improved methods for runoff interpolation. In this context, we believe that it is important to exploit all available data, also data from partially gauged catchments, which is what we call catchments

that only have short records of data, from a subset of the target period. Motivated by this, we propose how short records can be modeled in the spatially varying coefficient model together with data from fully gauged catchments. More specifically, we suggest to use the approach from Roksvåg et al. (2020) as a preprocessing technique for record augmentation for partially gauged catchments before further analysis with the spatially varying coefficient model. The approach from Roksvåg et al. (2020) is a geostatistical record augmentation procedure that is shown to work well for flow indices and study areas that are dominated by

runoff patterns that are repeated over time. Repeated runoff patterns are often seen for runoff observations of longer temporal scale, such as annual runoff, and particularly in areas where runoff is driven by constant factors such as topography, through e.g. orographic precipitation. In our proposed approach, the preprocessed short records are incorporated into the spatially varying





coefficient model through an observation likelihood that supports both data from fully gauged catchments and preprocessed data from partially gauged catchments. Differences in measurement uncertainties between the two data types are taken into
account through knowledge based prior distributions.

The main objective of this article is to present a framework for mean annual runoff interpolation that exploits several relevant data sources: Precipitation data, temperature data and land-use through the process-based covariate, and data from fully gauged and partially gauged catchments through the observation likelihood. The framework is made feasible by using state of the art statistical methods such as INLA and SPDE (Rue et al., 2009; Lindgren et al., 2011) that allows fast and approximate
inference for computationally expensive Bayesian methods. We evaluate the model by assessing the model's ability to produce a satisfactory gridded runoff map with corresponding uncertainty estimates and by evaluating the predictive performance of the method for fully gauged, partially gauged and ungauged catchments. For this purpose, we use mean annual runoff observations from Norway. Simulations of mean annual runoff produced by the process-based HBV model are used as a covariate and the HBV simulations are available on a 1 km × 1 km grid for the whole country. We compare our results to a purely process-based
reference model (the HBV model) and a purely geostatistical model (Top-Kriging).

In the next section (Section 2), we present the available Norwegian runoff data and model input. Here, we describe the process-based HBV model and how it was used to produce simulations on a grid. In Section 3 we introduce background theory, relevant statistical models and notation. Further, in Section 4, we step by step present the suggested mean annual runoff model, where the preprocessing step for including short records is described in Section 4.4. The experimental set-up for evaluating the
model is presented in Section 5, and in Section 6 and 7 we present and discuss our results. Finally, we summarize and conclude in Section 8.

## 2  Model input

### 2.1  Runoff data

To evaluate the proposed geostatistical approach, we use mean annual runoff data from Norway from the time period 1981-2010
provided by the Norwegian Water Resources and Energy Directorate (NVE). The mean annual runoff observations have unit mm/year and were derived by aggregating daily measurements of streamflow from Norwegian catchments, for hydrological years that starts September 1st and ends August 31st. If a catchment had less than 365 daily observations for a specific year, this annual observation was considered missing.

Furthermore, we only use data from catchments where human activities have had negligible impact. To select catchments,
we used the regulation capacity of hydropower reservoirs as a criterion, i.e. the ratio between the mean annual runoff and the reservoir storage capacity. If this ratio was smaller than 0.2 for a catchment, we assumed that the change in stored water could be ignored. Catchments with a ratio larger than 0.2 were omitted from the analysis.

After carrying out the data cleaning procedure explained above, there were data available from 127 catchments that were *fully gauged* in the 30 year target period, 1981-2010. Averaging these 30 years, gave the 127 streamflow measurements of mean
annual runoff that are shown in Figure 1a in mm/year. In addition, there were annual observations available from 284 additional





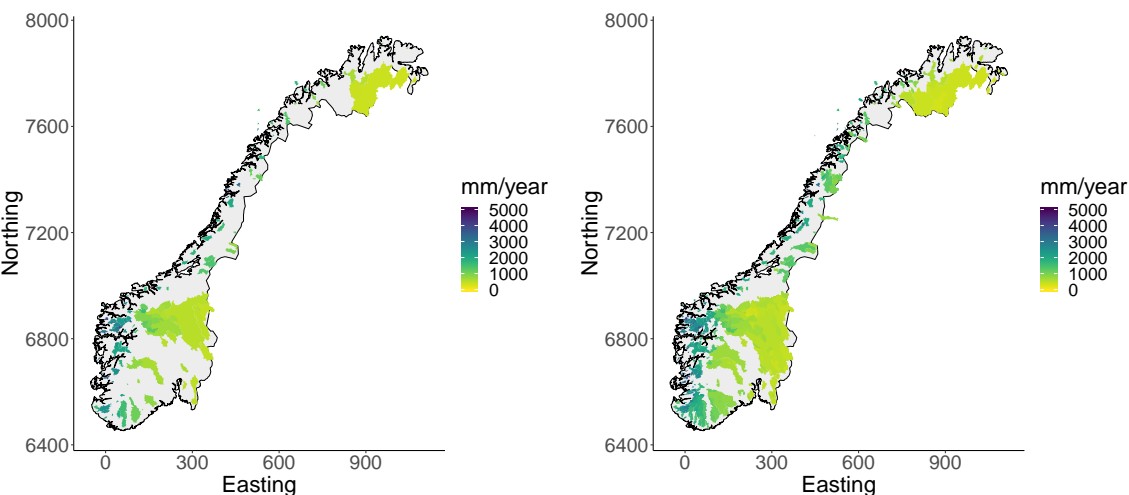

(a) Catchmens that are fully gauged in the study period (1981-2010).

(b) Fully and partially gauged catchments (1965-2010).

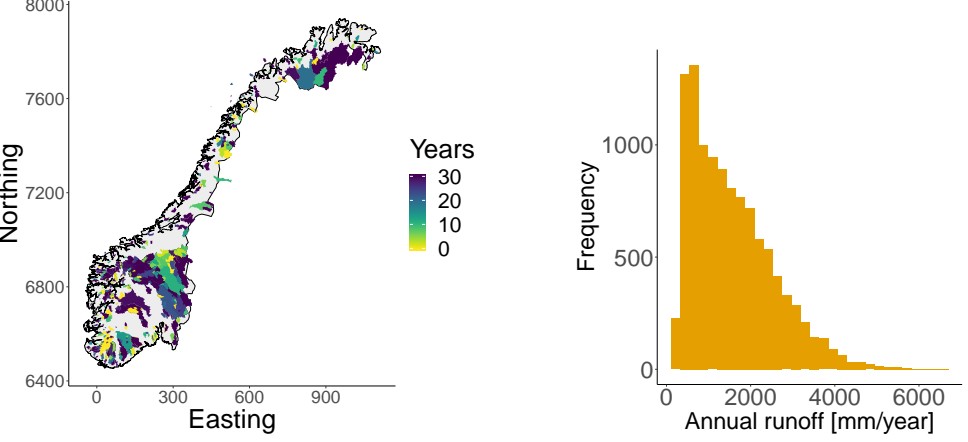

(c) Number of annual observations available for 1981-2010.

(d) Observed annual runoff from fully gauged and partially gauged catchments (1965-2010).

**Figure 1.** Mean annual runoff for Norwegian catchments (upper plots) derived from daily streamflow observations. There are annual runoff data available from 127 fully gauged catchments and from 284 partially gauged catchments. Many of the catchments in southern Norway are nested, and we plot subcatchments in front of larger, surrounding catchments in all of our plots. Figure 1c shows the number of annual observations that are available for each catchment in the study period (1981-2010). If this number is equal to 0, it means that there is at least one annual observation available from 1980 or earlier years, more specifically between 1965 and 1980. The reference system used is utm33N EUREF89 with coordinates given in km. Figure 1d shows the annual observations available for all catchments and years.





catchments. These were not fully gauged in the study period (1981-2010), but had at least 1 annual runoff observation between 1965 and 2010. We refer to these catchments as *partially gauged catchments* and their mean annual runoff is shown in Figure 1b. In Figure 1c we also show the number of annual observations available between 1981-2010 for each of these catchments. The average record length is 12 years (median 9.5 years) for 1981-2010, but 15 years (median 16 years) if we consider the

longer time period from 1965 to 2010.

In our analysis, we use the short runoff records from the partially gauged catchments from 1965-2010 to estimate the mean annual runoff for the same catchments for 1981-2010. This is done by applying a record augmentation preprocessing step before fitting the full SVC model presented in this article (see Section 4.4). In the preprocessing step, spatial interpolation is performed to fill in missing annual observations. The reason for including years before 1981 here, is that it makes it possible

to include more catchments in our analysis, i.e. catchments that only have data from before 1981.

In Figure 1d we show the annual runoff observations for individual years. Here, it is apparent that the spatial variability of the Norwegian annual runoff is large: It ranges from around 400 mm/year to around 6000 mm/year. The mean annual runoff follows the spatial pattern we see in Figure 1b, with large observations in the western part of the country and smaller observations in eastern part each year. The pattern is mainly caused by the orographic precipitation that occurs when humid

winds from the Atlantic ocean are elevated over the mountains in western Norway. This gives large precipitation amounts in the western parts of the country, while the eastern parts are left in the rain shadow.

## 2.2 Gridded simulations from the HBV model

We use a gridded mean annual runoff product simulated by the HBV model as a covariate in our geostatistical model. This was already available from the data provider NVE's database and is shown in Figure 2a. The gridded product was delivered on a

1 km × 1 km grid and is based on simulations of daily time series of runoff where interpolated temperature and precipitation were used as inputs together with land use characteristics. The daily simulated time series of runoff were aggregated to mean annual runoff (mm/year) for our reference period 1981-2010. We refer to Beldring et al. (2002) for details.

The HBV (Hydrologiska Byråns Vattenbalansmodell) model is a conceptual hydrological model that accounts for the key hydrological processes in a Nordic climate. The first application of the gridded HBV model in Norway is reported in Beldring

et al. (2002), and it is applied in several studies to assess runoff and water balance in Norway (e.g. in Borgvang et al. (2006); Skarbøvik et al. (2009); Hanssen-Bauer et al. (2017)). The HBV model is applied as a gridded model, typically on a daily time scale, and the water balance is estimated for each grid cell in a discretization of the study area, where the grid cells are characterized by elevation and land use. Different land use classes are associated with specific parameters that control the snow processes, interception storage, evapotranspiration and subsurface moisture storage, and runoff generation. Based

on the parameterization of these key hydrological processes, the HBV model sieve the precipitation into runoff (blue water) an evapotranspiration (green water). It is therefore the land use characteristics in each grid cell that controls the proportion of precipitation that generates runoff. We refer to Bergström (1976); Sælthun (1996); Lindström et al. (1997) for detailed descriptions of the algorithms used in the HBV model.





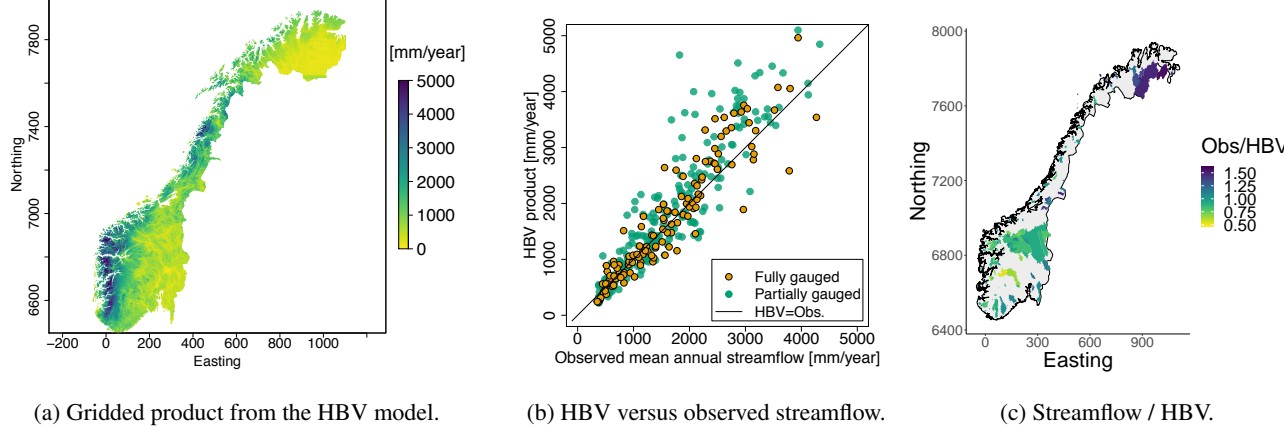

(a) Gridded product from the HBV model.     (b) HBV versus observed streamflow.     (c) Streamflow / HBV.

**Figure 2.** A mean annual runoff (1981-2010) product simulated by the HBV model (Figure 2a). The product is delivered on a 1 km × 1 km grid. Figure 2b shows the fit between the HBV product and the actually observed streamflow for the fully gauged (orange) and partially gauged catchments (green). In Figure 2c, the ratio between the observed streamflow and the HBV product is visualized spatially for the 127 fully gauged catchments from Figure 1a.

To determine the parameters in the HBV model, it is common to perform a global calibration procedure. The calibration
procedures performed on the product in Figure 2a are described in Beldring et al. (2003): Key parameters associated with each land use class were tuned, aiming at minimizing the global bias and the errors based on streamflow observations, and streamflow observations from 141 fully gauged catchments were used for the calibration. Mark that since we use the HBV product that already was available from the data provider NVE's database, the calibration catchments are not necessarily the same catchments we use in our geostatistical model. However, most of the 141 calibration catchments probably coincide with
the 127 fully gauged catchments in Figure 1a.

As the parameters of the HBV model are calibrated globally, there are still local biases in the runoff grid calculated by the HBV model relative to the observed streamflow. This can be seen in Figure 2b and 2c where we visualize the difference between the mean annual runoff provided by the HBV model and the actually observed mean annual runoff. The figures were produced by aggregating the gridded data in Figure 2a to the catchment areas in Figure 1b. Next, the average within each
catchment was plotted with the actually observed mean annual runoff.

In the scatter plot in Figure 2b, we see that the fit between the HBV model and the observed streamflow is close to linear for smaller observations of mean annual runoff. However, for observations over 2000 mm/year, the relationship is more skewed: The HBV model seems to overestimate the mean annual runoff for the most extreme values. By using the proposed geostatistical approach, we aim to produce a runoff map that improves the fit. In Figure 2c, the ratio between the observed mean annual
streamflow and the HBV product is visualized spatially. Here, it looks like there is some spatial trend where catchments that are located close in space have similar ratios. This is an indication that a spatially varying coefficient model is appropriate.





## 3 Methodological background

Before presenting the proposed runoff estimation approach, we briefly describe background theory and the statistical models we use to build our models.

### 3.1 Bayesian statistics and hierarchal modeling


When taking a statistical approach to hydrological modeling, the relationship between some observations $\boldsymbol{y} = (y_1,...,y_m)$ and the hydrological process of interest $\boldsymbol{x} = (x_1,...,x_n)$ is expressed through a statistical distribution, often through an observation likelihood which we denote by $\pi(\boldsymbol{y}|\boldsymbol{x})$. In this article, we take a Bayesian approach to statistics (see e.g.Gelman et al. (2004); Casella and Berger (1990)). This means that the random variable $\boldsymbol{x}$ is associated with a probability distribution that expresses

what we know about the underlying process of interest. Before the statistical analysis is conducted, our beliefs are expressed mathematically through a so-called prior distribution, denoted $\pi(\boldsymbol{x})$. The prior distribution can e.g. be constructed based on expert knowledge about the hydrological process under study or based on earlier experiments. The goal of the Bayesian analysis, is to update $\pi(\boldsymbol{x})$ based on data $\boldsymbol{y}$. This can be done by using Bayes' formula:

$$\pi(\boldsymbol{x}|\boldsymbol{y}) = \frac{\pi(\boldsymbol{x})\pi(\boldsymbol{y}|\boldsymbol{x})}{\pi(\boldsymbol{y})}, . \tag{1}$$

The resulting distribution $\pi(\boldsymbol{x}|\boldsymbol{y})$ is called the posterior distribution, and represents what we know about the underlying process *after* some evidence is taken into account, i.e. data. As our information about $\boldsymbol{x}$ is summarized through a statistical distribution, a full uncertainty specification is directly available. This is one of the benefits of taking a Bayesian approach. However, if a point prediction is of interest, the median, mean or mode of the posterior distribution $\pi(x_i|\boldsymbol{y})$ can be used as a summary statistic, for any $x_i \in \boldsymbol{x}$.

In this article, we present a Bayesian *hierarchical* model for mean annual runoff. Hierarchical modeling is a popular modeling framework as hierarchical models make it possible to formulate rather complex models by specifying a set of simpler models (see e.g. (Banerjee et al., 2004)). This is done in a hierarchical structure. For example if we are interested in modeling runoff, we can assume that the true underlying runoff $\boldsymbol{x}$ is observed through data $\boldsymbol{y}$ that are associated with some measurement uncertainty. Further can we assume that the runoff has some spatial or temporal variability that can be modeled by a statistical

distribution with parameters $\boldsymbol{\theta} = (\theta_1,...,\theta_k)$ . The parameters could e.g. be variance and correlation parameters, and as we use the Bayesian framework, the parameter vector $\boldsymbol{\theta}$ is as $\boldsymbol{x}$ associated with prior and posterior distributions. Mathematically, the above model can be expressed in a stage-wise manner: Here, the first level contains the *observation likelihood* $\pi(\boldsymbol{y}|\boldsymbol{x},\boldsymbol{\theta})$ that connects the data to the underlying processes. The second level is often referred to as the *latent model* or *process model* and contains the prior distribution $\pi(\boldsymbol{x}|\boldsymbol{\theta})$ of the random variables in $\boldsymbol{x}$ given the underlying parameters $\boldsymbol{\theta}$. The third level

contains the prior distribution of the model parameters $\pi(\boldsymbol{\theta})$. The goal of the Bayesian analysis, is to determine the posterior distributions of both $\boldsymbol{x}$ and $\boldsymbol{\theta}]$ given data $\boldsymbol{y}$ as before.



## 3.2 Gaussian random fields (GRFs)

Random fields (RFs) are often used to model spatial correlation in geostatistical models for hydrological variables (see e.g.
Sauquet et al. (2000); Skøien et al. (2006); Roksvåg et al. (2020)). In this article, we will use the most common class of
random fields to model runoff, namely Gaussian random fields (GRFs). A continuous field $\{x(\boldsymbol{u}); \boldsymbol{u} \in \mathcal{D}\}$ defined on a spatial
domain $\mathcal{D}$ is a Gaussian random field if $(x(\boldsymbol{u}_1), ..., x(\boldsymbol{u}_n))^{\mathrm{T}} \sim \mathcal{N}(\boldsymbol{\mu}, \boldsymbol{\Sigma})$ where $\mathcal{N}(\cdot, \cdot)$ is a multivariate normal distribution with
expected values given by vector $\boldsymbol{\mu}$ and covariance given by the covariance matrix $\boldsymbol{\Sigma}$ (Cressie, 1993). The covariance matrix
is central in spatial statistics as it specifies the dependency structure of the variable of interest. Often, a matrix element $(i, j)$
is generated by using a known covariance function $C(\boldsymbol{u}_i, \boldsymbol{u}_j)$ that models the correlation of the target variable between two
locations $\mathrm{Cov}(x(\boldsymbol{u}_i), x(\boldsymbol{u}_j)$. This covariance function typically has a marginal variance parameter $\sigma^2$ and a range parameter
$\rho$ that characterize the underlying spatial field: The marginal variance describes the spatial variability of the target variable,
while the range is a measure of how correlation decays with distance.

In our work we use a stationary Matérn covariance function to model the covariance of mean annual runoff. The Matérn
covariance function is defined as:

$$C(\boldsymbol{u}_i, \boldsymbol{u}_j) = \frac{\sigma^2}{2^{\nu-1}\Gamma(\nu)}(\kappa||\boldsymbol{u}_j - \boldsymbol{u}_i||)^\nu K_\nu(\kappa||\boldsymbol{u}_j - \boldsymbol{u}_i||), \tag{2}$$

where $K_\nu$ is the modified Bessel function of second kind and order $\nu > 0$, $\Gamma(\cdot)$ is the gamma function and $||\boldsymbol{u}_j - \boldsymbol{u}_i||$ is the
Euclidean distance between the two locations $\boldsymbol{u}_i, \boldsymbol{u}_j \in \mathcal{R}^d$. Further, is $\sigma^2$ the marginal variance and $\kappa$ is a scale parameter
(Guttorp and Gneiting, 2006). Empirically, it has been shown that the parameters $\nu$ and $\kappa$ can be used to express the spatial
range through the following relationship; $\rho = \sqrt{8\nu}/\kappa$, where $\rho$ is defined as the distance at which the correlation between two
locations has dropped to 0.1 (Rue et al., 2009).

As $\kappa$ and $\sigma^2$ are constant in Equation (2), the Matérn covariance model is stationary in space. The reason for using a Matérn
covariance function in our work, is that it comes with computational benefits: It makes it possible to use the SPDE approach
to spatial modeling from Lindgren et al. (2011) which is described in Section 4.6. In addition, the Matérn class of covariance
functions has many useful properties and Stein (1999) advice to use it.

## 230 3.3 Existing geostatistical models used for runoff interpolation

There exist several geostatistical models for interpolation of hydrological variables. In this work we refer to two of them: the
Top-Kriging approach from Skøien et al. (2006) and the geostatistical method for exploiting short records from Roksvåg et al.
(2020).

### 3.3.1 Top-Kriging

Kriging approaches are a set of approaches that can be used to predict spatial variables at unobserved locations. In Kriging
approaches, the variable of interest is modeled as a random field $x(\boldsymbol{u})$. An estimate of the random field $x(\boldsymbol{u}_0)$ at an unobserved



location $\boldsymbol{u}_0 \in \mathcal{R}^2$ can be expressed as the weighted sum of a set of observations $x(\boldsymbol{u}_i), ..., x(\boldsymbol{u}_n)$, i.e. as

$$\hat{x}(\boldsymbol{u}_0) = \sum_{i=1}^{n} \lambda_i x(\boldsymbol{u}_i), \tag{3}$$

where $\lambda_i$ for $i = 1, .. n$ are interpolation weights that must be determined (Cressie, 1993). The interpolation weights can be
specified by finding the set of weights that minimize the mean squared error between the estimate $\hat{x}(\boldsymbol{u}_0)$ and the true $x(\boldsymbol{u}_0)$, and that give zero mean expected error. A linear estimator with these properties is called the best linear unbiased estimator (BLUE).

The estimation of the Kriging weights requires evaluations of the covariance function of the involved random field and the covariance typically depends on the distance between the observation locations $\boldsymbol{u}_i$. However, runoff observations are
linked to catchment areas rather than to single point locations, and this should be taken into account when calculating the covariance. One way to do this is by using Top-Kriging Skøien et al. (2006), which is a a Kriging approach particularly suitable for interpolation of areal referenced hydrological variables. The method treats runoff observations as areal referenced in the covariance calculations and this way ensures that an observation from a subcatcment gets a higher Kriging weight than an observation from a nearby, non-overlapping catchment. According to Viglione et al. (2013); Blöschl et al. (2013) Top-Kriging
is one of the leading methods for interpolation of catchment referenced variables, and we hence use it as a geostatistical reference method in this article.

### 3.3.2    Geostatistical method for exploiting short records

In addition to Top-Kriging, we refer to the geostatistical method suggested in Roksvåg et al. (2020). This is a method particularly suitable for making predictions in catchments that have short records of data relative to their neighboring catchments.
The model in Roksvåg et al. (2020) is a Bayesian hierarchical geostatistical model that models several years of (annual) runoff simultaneously through two GRFs: one that describes the long-term spatial variability in the study area, and one that describes year dependent spatial effects. The method weights the two GRFs relative to each other and if long-term effects dominate, the potential information stored in short records is large.

In this paper, we use the method from Roksvåg et al. (2020) as a preprocessing step for making inference about the partially
gauged catchments in the dataset before further analysis with the spatially varying coefficient model. This way we increase the size of the dataset and are able to fully exploit all the available data in our study area, Norway. See Section 4.4 for further description of the preprocessing step.

The method from Roksvåg et al. (2020) has its benefits when modeling flow indices and study areas where there are characteristic spatial patterns of runoff that are repeated over time. This is the case for our target variable, Norwegian annual runoff,
that is driven by orographic precipitation caused by repeated wind patterns from the Atlantic ocean (Stohl et al., 2008). Example data are shown in 3, representing a setting for which record augmentation can have a large value. The repeated spatial pattern is recognized by that the time series are almost parallel over time, i.e. the ranking of the catchments, from wet to dry, is approximately constant. For variables and areas that are not driven by such characteristic spatial patterns, the method in Roksvåg et al. (2020) provides a more classical form of spatial interpolation, similar to Kriging methods.



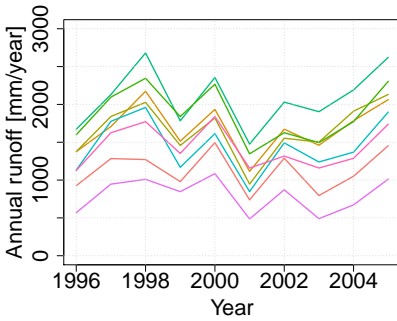

**Figure 3.** Time series of annual runoff for 8 catchments in Norway that are located relatively close to each other. The time series are almost parallel, indicating that the spatial patterns of runoff are repeated over time. This is a setting for which we can expect the method from Roksvåg et al. (2020) to contribute to increased predictive performance for partially gauged catchments.

The method from Roksvåg et al. (2020) is available for both point and areal referenced data. In this article we use it as a point referenced model to save computational time. The point referenced model uses the catchments centroids as the observation locations. Although an areal model is more realistic for runoff data, we expect the point referenced model to be sufficiently good for our area of use for two reasons: 1) We are only using the model to make predictions for catchments where we have at least one annual observation and 2) we are not going to use the posterior uncertainty of the model, as the final prediction uncertainty is determined by the spatially varying coefficient model. The results in Roksvåg et al. (2020) show that the point referenced model gives results that are approximately as good as the areal referenced model for partially gauged catchments when we only are interested in the posterior mean and not the posterior standard deviation.

## 4 A spatially varying coefficient (SVC) model for incorporating process-based simulations and short records

We now present a geostatistical Bayesian hierarchical model for mean annual runoff that incorporates simulations from a process-based model through a spatially varying coefficient and supports data from both fully gauged and partially gauged catchments. It is a three stage model that contains a process model, an observation model and prior distributions as outlined in Section 3.1.



### 4.1 Process model for true mean annual runoff

#### 4.1.1 Underlying point model

Assume that mean annual runoff (mm/year) is a continuous process that occurs for any point $\boldsymbol{u} \in \mathcal{R}^2$ in the landscape. We model the true mean annual runoff $q(\boldsymbol{u})$ at a point location or a (small) grid cell $\boldsymbol{u}$ as

$$q(\boldsymbol{u}) = \beta_0 + (\beta_1 + \alpha(\boldsymbol{u})) \cdot h(\boldsymbol{u}) + x(\boldsymbol{u}); \qquad \textbf{(SVC model)} \qquad (4)$$

$$x(\boldsymbol{u})|(\rho_x, \sigma_x) \sim \mathrm{GRF}(\rho_x, \sigma_x) \qquad\qquad \beta_0 \sim \mathcal{N}(0, (10000 \text{ mm/year})^2)$$

$$\alpha(\boldsymbol{u})|(\rho_\alpha, \sigma_\alpha) \sim \mathrm{GRF}(\rho_\alpha, \sigma_\alpha) \qquad\qquad \beta_1 \sim \mathcal{N}(0, 10000^2)$$

where $\beta_0$ is an intercept with a normal distributed prior. The variable $h(\boldsymbol{u})$ is a covariate that contains the simulated value generated by a process-based hydrological model at point location or a grid cell $\boldsymbol{u}$, and $(\beta_1 + \alpha(\boldsymbol{u}))$ defines a spatially varying coefficient (SVC). The spatially varying coefficient consists of one fixed effect $\beta_1$ and one component $\alpha(\boldsymbol{u})$ that changes in space. The spatial variability of $\alpha(\boldsymbol{u})$ is introduced by modeling it as a Matérn Gaussian random field given a range parameter $\rho_\alpha$ and a marginal standard deviation parameter $\sigma_\alpha$. This way the relationship between the true mean annual runoff $q(\boldsymbol{u})$ and

the simulations made by the hydrological model $h(\boldsymbol{u})$ is allowed to vary in the study area. For the fixed effect $\beta_1$ we use a weakly informative normal prior distribution with zero mean and standard deviation 10000. This is the same prior as for the intercept $\beta_0$. In our application, we use mean annual runoff simulations from the HBV model in $h(\boldsymbol{u})$, but gridded simulations from any relevant hydrological model can be used as input.

The spatial dependency structure introduced by the spatially varying coefficient $\beta_1 + \alpha(\boldsymbol{u})$ in Equation (4) models a similar
dependency structure as we would obtain from performing *ratio interpolation*, i.e. interpolation of the ratio between the observed runoff and a process-based covariate. Ratio interpolation is a method that has been used before in e.g. Beldring et al. (2002) to improve the results from a process-based model. However, in our model, we also include an additional spatial effect $x(\boldsymbol{u})$ as we see in Equation (4). Like $\alpha(\boldsymbol{u})$, $x(\boldsymbol{u})$ is modeled as a GRF with Matérn covariance, but with range and marginal standard deviation $\rho_x$ and $\sigma_x$ respectively. The GRF $x(\boldsymbol{u})$ models a different dependency structure than $\alpha(\boldsymbol{u})$, more specifically
a dependency structure similar to what we would obtain from performing *residual interpolation*. Residual interpolation was used in e.g. Merz and Blöschl (2005) to improve the results from an initial multiple linear regression model.

The motivation behind including two spatial fields in our SVC model, is that it introduces flexibility to both the mean and the standard deviation of the predicted mean annual runoff, and makes it possible to model underlying processes with both long and short spatial ranges or large and small variances. Furthermore can the model itself detect which of the two dependency
structures that are most prominent in the data and adjust the spatial components relative to each other.

#### 4.1.2 Areal model

In Equation (4) we modeled runoff as a point referenced process. However, in practice, runoff is observed through streamflow observations that are linked to catchment *areas*. Because of this, we now introduce a model for the true mean annual runoff





inside a catchment area $\mathcal{A}$. This is given by:

$$Q(\mathcal{A}) = \frac{1}{|\mathcal{A}|} \int_{\boldsymbol{u} \in \mathcal{A}} q(\boldsymbol{u}) d\boldsymbol{u}, \tag{5}$$

where $q(\boldsymbol{u})$ is the mean annual point runoff from Equation (4) and $|\mathcal{A}|$ is the area of the target catchment. Hence, the true areal runoff is here given by the average point runoff integrated over the catchment area. However, in practice, it is not computationally feasible to perform the integration in Equation (5). The solution to this problem is to approximate the integral in Equation (5) by a sum. This is done by discretizing catchment $\mathcal{A}$ to a regular grid $\mathcal{L}_{\mathcal{A}}$ and defining the mean annual runoff in catchment

$\mathcal{A}$ as:

$$Q(\mathcal{A}) \approx \frac{1}{n_{\mathcal{A}}} \sum_{\boldsymbol{u} \in \mathcal{L}_{\mathcal{A}}} q(\boldsymbol{u}), \tag{6}$$

where $n_{\mathcal{A}}$ is the number of grid nodes in the discretization of catchment $\mathcal{A}$.

We have now defined our final process model for runoff. This is an areal model (Equation (6)) that builds on a point specification of the underlying process (Equation (4)). Comparing equations (4) and (6), we see that calculating $Q(\mathcal{A})$ requires

an evaluation of the quantity $h(\mathcal{A}) = \sum_{\boldsymbol{u} \in \mathcal{L}_{\mathcal{A}}} h(\boldsymbol{u})$, i.e. we have to aggregate the simulated values produced by the hydrological product $h(\boldsymbol{u})$ for the grid nodes inside $\mathcal{A}$. Consequently, the catchment discretization should follow the same discretization as the gridded hydrological product that is used as input to $h(\boldsymbol{u})$. In our case the HBV product comes on a regular grid with 1 km spacing. Mark that the grid should be dense enough to ensure an accurate approximation for the true areal runoff in Equation (6) and to avoid unrealistic results such as negative runoff.

## 4.2 Observation model for mean annual runoff

The true mean annual $Q(\mathcal{A})$ runoff is not observed directly, but through areal referenced streamflow observations with uncertainty. The observed mean annual runoff in catchment $\mathcal{A}_i$ is here modeled as:

$$y_i = Q(\mathcal{A}_i) + \epsilon_i, \tag{7}$$

where $Q(\mathcal{A}_i)$ is the areal referenced true mean annual runoff from Equation (6), and the $\epsilon_i$'s are independent and identically

distributed error terms with prior $\mathcal{N}(0, s_i \cdot \sigma_y^2)$. Here $\sigma_y$ is a parameter describing the underlying standard deviation, while the $s_i$'s are fixed, predetermined scales that allow each observation to have its own measurement uncertainty. This way heteroscedasticity can be introduced in a simple way. The values of the scales $s_i$ are further specified in Section 4.3.

It is convenient to use the areal formulation from Equation (6) to model the observed mean annual runoff. The reason is that it allows us to constrain the mean annual runoff not only at certain gauging points, but over the whole catchment area of

the gauged catchments. However, bear in mind that the constraints imposed by the likelihood only work as soft constraints. This means that the actually observed mean annual runoff over a catchment area is not guaranteed to be reproduced in the final predictive model. Whether the actually observed mean annual runoff is reproduced depends on e.g. the observation uncertainty $s_i \sigma_y^2$.



### 4.3  Prior distributions for model parameters

The third stage of the proposed hierarchical model for mean annual runoff consists of the prior distributions of the 5 model
parameters, $(\rho_\alpha, \sigma_\alpha, \rho_x, \sigma_x, \sigma_y)$. In this section we specify the prior distributions we use in our experiments. Most of these
priors are constructed such that they are suitable for Norwegian mean annual runoff data, and should be revised before the
model is used for other flow indices and/or study areas.

We start by constructing a prior for the measurement uncertainty expressed by $s_i \sigma_y^2$. As stated in the previous subsection,
the variance parameter $\sigma_y^2$ is scaled with a fixed an predetermined scale $s_i$ such that each observation of mean annual runoff
can have its own measurement uncertainty. A variance that changes with the observed value is reasonable when modeling
Norwegian mean annual runoff, because Norway is a diverse country when it comes to runoff generation: Most observations are
between 500 mm/year and 4000 mm/year. With this in mind, we specify the scales $s_i$ such that the measurement uncertainties
depend on the magnitude of the observed value, i.e. we assume that larger observations of mean annual runoff have larger
measurement uncertainties than smaller observations of mean annual runoff. This is obtained by modeling the scales as

$$s_i = (0.025 \cdot y_i/1000)^2, \tag{8}$$

where $y_i$ is the observed mean annual runoff in catchment $\mathcal{A}_i$ in mm/year. The number 0.025 was chosen according to expert
opinions from the data provider NVE: A standard deviation around 2.5 % is assumed to be reasonable. Further, are the scales
rescaled by a factor of 1000 to get suitable values for the quantity $s_i \cdot \sigma_y^2$.

Next, we need to specify a prior distribution for the standard deviation parameter $\sigma_y$. For this parameter, we use a penalized
complexity (PC) prior as suggested by Simpson et al. (2017). The PC prior is chosen because it has convenient mathematical
properties. It controls for overfitting by penalizing the increased complexity that arises when a more flexible model deviates
from a simpler base model. The PC prior for the precision $\tau$ (or the inverse variance) of a Gaussian effect $\mathcal{N}(0, \tau^{-1})$ is given
by

$$\pi(\tau) = \frac{\lambda}{2} \tau^{-3/2} \exp(-\lambda \tau^{-1/2}), \qquad \tau > 0, \quad \lambda > 0, \tag{9}$$

where $\lambda$ controls the deviation penalty. The parameter $\lambda$ can easily be specified through a probability $\alpha$ and a quantile $u$ as
$\text{Prob}(\sigma > \sigma_0) = \alpha$, where $\sigma_0 > 0$, $0 < \alpha < 1$ and $\lambda = -\ln(\alpha)/u$, where $\sigma = 1/\sqrt{\tau}$ is the standard deviation of the Gaussian
effect. For our application, we let $\alpha = 0.1$ and $\sigma_0 = 1500$ mm /year, and determine the PC prior for $\sigma_y$ as follows:

$$\text{Prob}(\sigma_y > 1500 \ \ \text{mm}) = 0.1. \tag{10}$$

This means that the prior probability that $\sigma_y$ is larger than 1500 mm/year is 10 %. However, recall that the measurement
variance of $y_i$ is determined by $s_i \sigma_y^2$ and not by $\sigma_y^2$ alone. With the scales in Equation (8) and the PC prior for $\sigma_y$ in Equa-
tion (10), a prior 95% credibility interval for the observation standard deviation for the mean annual runoff is $(0.04, 6)\%$ of
the corresponding observed value $y_i$ for a catchment $\mathcal{A}_i$, with the prior mean centered around 2.5%. Values in this range are
reasonable and reflect the data provider NVE assumptions about the uncertainty of the Norwegian mean annual runoff obser-





vations. Furthermore, we also want a quite narrow prior credible interval for $s_i\sigma_y^2$ in this context: This way we influence the model to reproduce the actually observed runoff in catchments where we have data, through the likelihood in Equation (7).

In Fuglstad et al. (2019) the PC prior framework is used to develop a knowledge-based, joint prior for the range and the marginal variance of a Gaussian random field. We use this prior for constructing a joint prior distribution for the spatial marginal standard deviation $\sigma_\alpha$ and the spatial range $\rho_\alpha$ for the spatially varying coefficient component $\alpha(\boldsymbol{u})$. The prior is

specified through the following probabilities and quantiles

$$\text{Prob}(\rho_\alpha < 20 \text{ km}) = 0.1, \qquad \text{Prob}(\sigma_\alpha > 2) = 0.1, \tag{11}$$

where we a priori assume that the spatial range of the spatially varying coefficient is larger than 20 km. This is a reasonable assumption for a study area that is approximately 40 km from west to east on its widest, and around 1600 km from north to south. Based on Figure 2b and Figure 2c we also assume a prior that the ratio between the response variable $Q(\cdot)$ and the

covariate $h(\cdot)$ varies with a factor that has a standard deviation smaller than 2.

Likewise, we use the PC prior from Fuglstad et al. (2019) to specify a joint prior for the marginal standard deviation $\sigma_x$ and the spatial range $\rho_x$ of the spatial effect $x(\boldsymbol{u})$. We use the following probabilities and quantiles:

$$\text{Prob}(\rho_x < 20 \text{ km}) = 0.1, \qquad \text{Prob}(\sigma_x > 2000 \text{ mm/year}) = 0.1. \tag{12}$$

Here, we again assume a prior that the range is larger than 20 km by taking the size of the study area into account. The

prior probability that the standard deviation of the Norwegian mean annual runoff is larger than 2000 mm/year is set to a low probability. This sounds reasonable as most of the mean annual observations are between 500 mm/year and 4000 mm/year.

### 4.4 Preprocessing step for incorporating short records (PP)

We now present an extension of the model that makes it possible to include short records in the analysis of mean annual runoff. For catchments that are fully gauged in the time period of interest, observations $y_i$ of mean annual runoff are directly

available. However, often there are also short records of data available from partially gauged catchments that only have annual observations available from a subset of the target period or from years before the target period. To incorporate the latter in our geostatistical model, we use the geostatistical model described in Section 3.3.2 as a preprocessing step for partially gauged catchments. The preprocessing step is used to fill in missing annual runoff observations for the partially gauged catchments in order to get better approximations of the mean annual runoff here. For this purpose, the observations from 1965-2010 from

Figure 1b are used. Next, the predictions of mean annual runoff (posterior mean) for 1981-2010 obtained from this approach are used as observations $y_i$ in the observation likelihood in Equation (7), together with data from fully gauged catchments.

The data $y_i$ we obtain from the preprocessing step are probably more uncertain than the data from the fully gauged catchments. To reflect this, we use a different prior observation uncertainty for the preprocessed data than for the fully gauged catchments' data. Recall that the prior observation variance for a fully gauged catchment was given by $s_i\sigma_y^2$ where $s_i$ was a

fixed predetermined scale given by $s_i = (0.025 \cdot y_i/1000)$. For partially gauged catchments we replace this scale by

$$s_i^{\text{PP}} = (0.10 \cdot y_i/1000), \tag{13}$$





where PP denotes that the observation $y_i$ from catchment $A_i$ is preprocessed. In practice, each partially gauged catchment could have its own scaling factor, but in this demonstration we use the same scaling factor for all partially gauged catchments for simplicity. With the scales in Equation (13), a 95 % credible interval for the prior standard deviation $\sqrt{s_i \sigma_y^2}$ becomes

$(0.1, 24)$ % of the observed value for the partially gauged catchments, while it is only $(0.04, 6)$ % for data from fully gauged catchments.

By including the preprocessing step, we have the possibility to exploit streamflow observations from catchments that have down to one annual observation. As explained in Section 3.3.2 the preprocessing step should work well for study areas and flow indices that are driven by repeated spatial patterns over time. If this is not the case, the preprocessing step only performs

classical geostatistical spatial interpolation and can be skipped to save computational time.

### 4.5  Full model specification

We have proposed a model for mean annual runoff that can incorporate process-based simulations, data from fully gauged and partially gauged catchments. We can now specify the full, Bayesian, hierarchical model for mean annual runoff as follows:

$$\pi(\boldsymbol{y}|\boldsymbol{x}, \sigma_y) \sim \prod_{i=1}^{n} (I\{\text{Catchment } A_i \text{ is fully gauged}\} \cdot \mathcal{N}(Q(A_i), s_i \sigma_y^2) +$$

$$I\{\text{Catchment } A_i \text{ is partially gauged}\} \cdot \mathcal{N}(Q(A_i), s_i^{\text{PP}} \sigma_y^2) \quad \text{[Observation likelihood]}$$

$$\pi(\boldsymbol{x}|\boldsymbol{\theta}) = \pi\left(x(\boldsymbol{u_1}), ..., x(\boldsymbol{u_m})|\rho_x, \sigma_x\right)$$
$$\cdot \pi\left(\alpha(\boldsymbol{u_1}), ..., \alpha(\boldsymbol{u_m})|\rho_\alpha, \sigma_\alpha\right) \cdot \pi(\beta_0) \cdot \pi(\beta_1) \quad \text{[Latent Model]} \tag{14}$$

$$\pi(\sigma_y, \boldsymbol{\theta}) = \pi(\rho_x, \sigma_x) \cdot \pi(\rho_\alpha, \sigma_\alpha) \cdot \pi(\sigma_y). \quad \text{[Prior]}$$

Here, $\boldsymbol{y}$ is a vector containing all observations $y_i, ..., y_n$ of mean annual runoff for catchments $A_1, ..., A_n$. The function $I(\cdot)$ is an indicator function that is equal to one if its argument is true and zero otherwise, allowing for data from both fully gauged and partially gauged catchments. Mark that the likelihood specification for the fully and partially gauged catchments is the

same, except for the difference in measurement uncertainty expressed through the predetermined scales $s_i$ and $s_i^{\text{PP}}$. Further is the variable $\boldsymbol{x}$ a vector that contains all the latent variables, i.e. the two fixed effects $\beta_0$ and $\beta_1$, and the two Gaussian random fields $x(\boldsymbol{u_1}), ..., x(\boldsymbol{u_m})$ and $\alpha(\boldsymbol{u_1}), ..., \alpha(\boldsymbol{u_m})$ for all grid nodes $\boldsymbol{u_1}, .., \boldsymbol{u_m}$ that are used in the discretization of the catchment areas. Finally is $\boldsymbol{\theta}$ a parameter vector that contains $\rho_x, \sigma_x, \rho_\alpha$ and $\sigma_\alpha$. Together with $\sigma_y$ it defines all the model parameters.

In Figure 4 we visualize the proposed approach in a flow chart. We emphasize that the SVC model can be used with

or without incorporating preprocessed short records. To mark results where preprocessed data are involved, we will use the subscript PP in the remainder of this text.





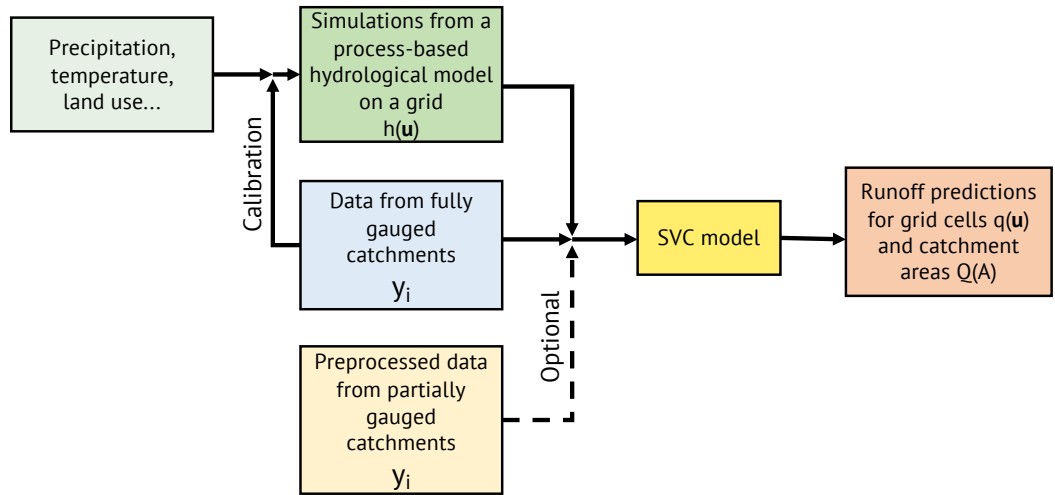

**Figure 4.** Workflow for runoff predictions on a grid $q(\boldsymbol{u})$ and for catchment areas $Q(\mathcal{A})$.

## 4.6 Approximate inference

The goal of Bayesian inference is to estimate the posterior distributions of the variables and parameters of interest, as described in Section 3.1. In this case we need to estimate $\boldsymbol{x}$ and $\boldsymbol{\theta}$ given data $\boldsymbol{y}$ in order to predict the mean annual runoff $q(\boldsymbol{u})$ and $Q(\mathcal{A})$

for our grid cells $\boldsymbol{u}$ and catchments $\mathcal{A}$. Traditionally, inference on hierarchical models of this type has been done by using Markov Chain Monte Carlo (MCMC) methods (Gamerman and Lopes, 2006). However, when considering mean annual runoff for the whole country of Norway, the dimension of the vector of latent variables $\boldsymbol{x}$ is large and hence also the computational complexity of carrying out a MCMC procedure. To solve this problem and make the model computationally feasible, integrated nested Laplace approximations (INLA) are used. The INLA methodology was suggested by Rue et al. (2009) and can be used

for making approximate Bayesian inference on latent Gaussian models (LGMs), i.e. hierarchical models where the latent field $\boldsymbol{x}$ is Gaussian. As the latent variables contained in $\boldsymbol{x}$ are given Gaussian prior distributions given the model parameters, this requirement is fulfilled for our SVC model. The INLA methodology is based on Laplace approximations, sparse matrix calculations and numerical integration schemes, and we refer to Rue et al. (2009) for details.

Furthermore, it is also computationally challenges related to performing matrix operations on the covariance matrices of

GRFs when we have many target locations, and our model contains not only one GRF, but two. To solve this issue, we use the SPDE approach to spatial modeling, as suggested by Lindgren et al. (2011). The approach is based on the fact that a GRF with Matérn covariance matrix can be expressed as the solution of a stochastic partial differential equation (Whittle, 1954, 1963). An approximate solution of the SPDE can be obtained by using the finite element method (see e.g. Brenner and Scott (2008)), where the resulting approximation is given on a triangular mesh. This mesh approximation has computational benefits

compared to the exact GRF solution. This enables fast inference for spatial models (Rue and Held, 2005; Rue et al., 2009).



The INLA and SPDE methodology is implemented in the r-package `INLA` and has since its introduction been used within a range of different fields. See Opitz et al. (2018); Guillot et al. (2014); Myrvoll-Nilsen et al. (2020); Bakka et al. (2018); Blangiardo and Cameletti (2015); Khan and Warner (2018) and `www.r-inla.org` for some examples. The approximations used in the SPDE and INLA framework are in general accurate and reliable when the likelihood is Gaussian, as in this application,

and as long as the triangular mesh used in the finite element computations is dense enough relative to the spatial variability of the target variable. A mesh that is too coarse can in our application lead to unrealistic results such as negative runoff.

## 5 Experimental set-up and evaluation scores

### 5.1 Experimental set-up

The goal of the article is to present and evaluate the geostatistical framework that incorporates process-based simulations and
short records. We evaluate the proposed approach in terms of making a gridded mean annual runoff map that improves the original HBV map in areas where we have observations, and in terms of performing accurate predictions for ungauged and partially gauged catchments. These two evaluation settings are described in Section 5.1.1 and 5.1.2 respectively.

### 5.1.1 Making a gridded mean annual runoff map for 1981-2010

To evaluate the proposed approach for runoff estimation, we fit the SVC model described in Section 4 to the Norwegian
mean annual runoff data. The observations in Figure 1b are used. These are observations from 127 fully gauged catchments from 1981-2010 and 284 partially gauged catchments from 1965-2010. For the partially gauged catchments, the preprocessing step (PP) described in Section 4.4 is performed before further analysis such that short records can be incorporated into the observation likelihood. The workflow is hence as visualized in Figure 4.

The result of the above procedure is a mean annual runoff map for 1981-2010 on the same grid as the HBV model in Figure
2a. We evaluate whether the new map improves the original HBV map by investigating how well the new map fits with the actually observed runoff from the fully gauged and partially gauged catchments.

In addition to the experiment described above, we repeated the experiment when omitting partially gauged catchments and short records from the analysis. This was done to show that the SVC model works regardless of the preprocessing step. These results can be found in Appendix A.

### 5.1.2 Cross-validation for ungauged and partially gauged catchments

We next assess the framework's ability to perform accurate mean annual runoff for ungauged and partially gauged catchments. This is done by a cross-validation assessment of the 127 fully gauged catchments from Figure 1a. The 127 fully gauged catchments are divided into five groups or folds: The four first folds have 25 so-called target catchments, while the fifth fold has 27 target catchments. The cross-validation folds are rather large because of the computational complexity of the problem.
In turn, the streamflow data corresponding to each fold are removed from the dataset, while the remaining observations are used





to predict the mean annual runoff for these catchments for 1981-2010. The likelihood consists of preprocessed observations from partially gauged catchments and observations from fully gauged catchments, i.e. around 400 observation catchments in total. Hence, the workflow is as in Figure 4. However, mark that we don't calibrate the HBV model for each cross-validation fold: The HBV product was a pre-made product available in the data provider's database, and we use the same HBV product

for all experiments without any modifications.

In our evaluation, we compare the predictive performance of the **SVC model** with the process-based **HBV model**. Hence, the original simulations from the HBV model shown in Figure 2a are used as they are as process-based reference predictions. For evaluation purposes, the values in Figure 2a are aggregated and averaged to catchment runoff for the catchments in Figure 1.

We also compare our approach to the purely geostatistical **Top-Kriging (TK)** approach. Top-Kriging is used to predict mean annual runoff in ungauged catchments based on a weighted sum of observations from nearby catchments, as described in Section 3.3. For this purpose, a covariance model based on a multiplication of a modified exponential and fractal variogram model is fitted to the data. This was used because it was the default option in the R package `rtop`. As for the SVC model, data from both fully gauged catchments and preprocessed partially gauged catchments are used for Top-Kriging, and we mark

the Top-Kriging results by TK$_{PP}$ to emphasize that preprocessed data are used as input. For fully gauged catchments, the standard deviation of the observations is set to 2.5 % of the observed value $y_i$ in the Top-Kriging approach. For partially gauged catchments the standard deviation is set to 10 % of the observed value. This is done to make the results as comparable as possible to our proposed SVC model.

In addition to evaluating Top-Kriging and the HBV model, we also include prediction results from the **preprocessing step**

**(PP)** alone, without performing any further analysis. The PP predictions come from the purely geostatistical method described in Section 3.3.2. We include the PP results to make the Top-Kriging and SVC results more transparent: These methods use the PP results as input data, representing observations from partially gauged catchments (see Section 4.4).

The described cross-validation procedure is first performed when the 127 target catchments are treated as ungauged. Hence, we have the following setting:

**Ungauged catchments (UG):** The target catchments in each cross-validation fold are treated as totally ungauged (UG) in the time period of interest (1981-2010) and their observations are removed from the dataset. Observations from fully gauged catchments (1981-2010) from other cross-validation groups and observations from partially gauged catchments (1965-2010) are used to make predictions.

We also evaluate the predictive performance of the model when the 127 target catchments are treated as partially gauged by doing the following experiment:

**Partially gauged catchments (PG):** The target catchments in each cross-validation fold are treated as partially gauged (PG). By this we mean that each target catchment is allowed to have a few annual observations in the study period (1981-2010), in this case 3 annual observations. These are drawn randomly from 1981-2010 for each target catchment. The remaining 27

observations from the target catchment are removed (and observations from before 1981). The preprocessing step from Section 4.4 is used to make inference about the mean annual runoff for the target catchments before using these as observed values





in the SVC model or Top-Kriging. In addition, observations from nearby fully gauged catchments (1981-2010) and partially gauged neighboring catchments (1965-2010) are included in the likelihood as before.

The same cross-validation groups are used for all experiments, such that the results become comparable across methods. The randomly drawn short records of length 3 are also the same for Top-Kriging and the SVC approach.

    In addition to the above experiments, we carried out a cross-validation for the UG setting when omitting catchments with short records and preprocessed data. These results can be found in Appendix A.

## 5.2   Evaluation scores

To evaluate the accuracy of the predictions obtained from the cross-validation, we use three evaluation scores. These are the root mean square error (RMSE), the absolute normalized error (ANE) and the Nash-Sutcliffe model efficiency coefficient (NSE), which are defined as:

$$\text{RMSE} = \sqrt{\frac{1}{n}\sum_{i=1}^{n}(y_i - \hat{Q}(\mathcal{A}_i))^2}, \tag{15}$$

$$\text{ANE}_i = \frac{|y_i - \hat{Q}(\mathcal{A}_i)|}{y_i}, \tag{16}$$

and

$$\text{NSE} = 1 - \frac{\sum_{i=1}^{n}(\hat{Q}(\mathcal{A}_i) - y_i)^2}{\sum_{i=1}^{n}(y_i - \overline{y})^2}. \tag{17}$$

Here, $\hat{Q}(\mathcal{A}_i)$ is the predicted mean annual runoff in catchment $\mathcal{A}_i$, $y_i$ is the corresponding observed value and $\overline{y}$ denotes the average observed mean annual runoff over all study catchments $i = 1, ... n$. For the suggested SVC model, we use the posterior

mean of $Q(\mathcal{A})$ as the predicted value (Equation (6)). As a summary statistic for $\text{ANE}_i$, we use the average $\text{ANE}_i$ over all catchments $i = 1, .. n$. A low average $\text{ANE}_i$ or a low RMSE corresponds to accurate predictions. The NSE on the other hand takes values between $-\infty$ and 1, and the closer the model efficiency is to 1, the more accurate the model is. The ANE and the NSE are different from the RMSE in being scale-independent evaluation scores.

    The three above scores are suitable for evaluating prediction bias, but they do not evaluate the models' uncertainty quantifi-

cation. For this reason we introduce two additional evaluation scores: the continuous ranked probability score (CRPS) and the 90 % coverage. The CRPS is in general given by

$$\text{CRPS}(F, y) = \int_{-\infty}^{\infty} (F(s) - 1\{y \leq s\})^2 ds,$$

where $y$ is the observed value and $F(\cdot)$ is the predictive cumulative distribution (Gneiting and Raftery, 2007). From the above definition, we see that CRPS takes the whole posterior distribution $F(\cdot)$ into account, unlike RMSE, ANE and NSE that only





consider point predictions. A low CRPS corresponds to an accurate prediction, and the CRPS increases if the observed value $y$ falls outside the posterior predictive distribution $F(\cdot)$. In this application, we assume $F(\cdot)$ to be Gaussian distributed with expected value given by the predicted mean annual runoff and standard deviation equal to the corresponding predictive standard deviation. The Gaussian assumption should be reasonable, as the posterior distributions of the predicted runoff typically are symmetric with light tails. We use the average CRPS over the 127 fully gauged catchments as a summary score.

Finally, the 90 % coverage is defined as the probability that 90 % of the observed values are covered by the corresponding 90 % posterior prediction intervals. This probability is computed empirically based on the predictions for the 127 fully gauged catchments, assuming that the SVC and Top-Kriging predictions follow a Gaussian distribution. If the empirical probability is close to 0.9 for a model, it suggests that the model provides an appropriate uncertainty quantification for the underlying variable.

## 6    Results

In Section 6.1 we present the gridded mean annual runoff map obtained from the experiment described in Section 5.1.1. Next, in Section 6.2, we present the results from the cross-validation described in Section 5.1.2. Together, the two experiments show how the suggested framework performs for fully gauged, partially gauged and ungauged catchments in Norway.

### 6.1    Gridded mean annual runoff map for 1981-2010

In Figure 5a we present the runoff map produced by the SVC$_{PP}$ approach. The difference between the new map and the original HBV product is visualized in Figure 5b, while the map's uncertainty estimates are shown in Figure 5c. Recall that the subscript PP refers to that the method uses preprocessed data. Figure 5b shows that the SVC$_{PP}$ map gives lower values of mean annual runoff in western Norway compared to the original HBV map. The difference is around 700-1500 mm/year. In eastern Norway, the original HBV map and the SVC$_{PP}$ maps are approximately equal, both in south-east and north-east. Around the glacier called *Svartisen*, located in northern Norway in the area where Norway is most narrow, the mean annual runoff of the SVC$_{PP}$ map is lower than the mean annual runoff of the original HBV map with a difference around 1500 mm/year.

We see from Figure 5a that the SVC$_{PP}$ map preserves most of the details provided by the original gridded HBV product in Figure 2a. The runoff map produced by SVC$_{PP}$ also looks visually good without e.g. unrealistic jumps or obvious discontinuities. One exception is a line or discontinuity close to the Finnish border, north-east in Figure 5a, but this line was already present in the original HBV product in Figure 2a.

The reason that most of the details from the original HBV map are preserved, is that the covariate $h(\boldsymbol{u})$ makes a large contribution to the final model with a regression coefficient $\beta_1$ that is estimated to be 0.83. This can be seen in Table 1 where we present the parameter estimates of the SVC model. In Table 1 we also see that the marginal standard deviations $\sigma_\alpha$ and $\sigma_x$ of the two spatial fields $\alpha(\boldsymbol{u})$ and $x(\boldsymbol{u})$ are significant in magnitude, confirming that there indeed is a regional trend in the fit between the original HBV product and the actually observed mean annual runoff. The regional trend can be studied in Figure 6 where we have included a visualization of the two spatial fields $\alpha(\boldsymbol{u})$ and $x(\boldsymbol{u})$. We see that the spatial pattern





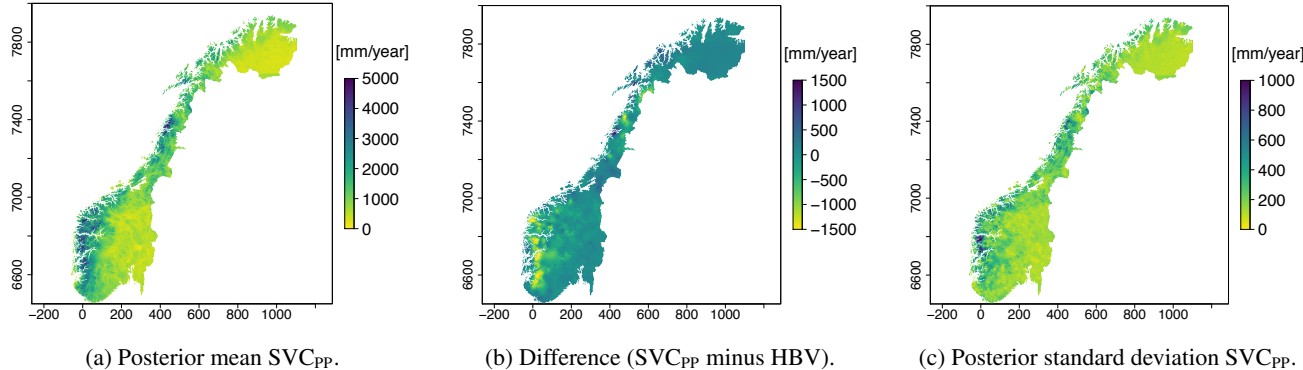

(a) Posterior mean SVC$_{PP}$.    (b) Difference (SVC$_{PP}$ minus HBV).    (c) Posterior standard deviation SVC$_{PP}$.

**Figure 5.** Posterior mean of $q(\boldsymbol{u})$ for all grid nodes $\boldsymbol{u}$, difference between the new map and the original HBV map and posterior standard deviation of $q(\boldsymbol{u})$.

**Table 1.** Posterior median (0.025 quantile , 0.975 quantile) for the parameters of the SVC$_{PP}$ model.

| Parameter [unit] | SVC$_{PP}$ |
|---|---|
| $\beta_0$ [mm/year] | 153 (110,196) |
| $\beta_1$ [1] | 0.83 (0.78,0.90) |
| $\rho_x$ [km] | 10.7 (5.4,26.1) |
| $\sigma_x$ [mm/year] | 117 (33.8,292) |
| $\rho_\alpha$ [km] | 39.2 (29.4,51.9) |
| $\sigma_\alpha$ [1] | 0.24 (0.21,0.27) |
| $\sigma_y$ [mm/year] | 205 (177,1000) |

in Figure 5a mostly originates from the spatially varying coefficient component $\alpha(\boldsymbol{u})$ for SVC$_{PP}$ (Figure 6a). The other GRF $x(\boldsymbol{u})$ contributes with more local adjustments in the mean annual runoff (Figure 6b). Hence, the spatial fields have picked up both short ranged and long ranged processes.

Next, considering the posterior standard deviation of the SVC$_{PP}$ model in Figure 5c, we see two trends: (i) The model gives a posterior uncertainty that follows the pattern we see in the original HBV map in Figure 2a and (ii) if we look closely at Figure 5c, we see that the uncertainty is decreased in areas where there are observations, particularly around the centroids of the gauged catchments. Here, it is the spatially varying coefficient $(\beta_1 + \alpha(\boldsymbol{u})) \cdot h(\boldsymbol{u})$ from Equation (4) that causes pattern (i). Including only the GRF $x(\boldsymbol{u})$ would only give pattern (ii). Figure 5c further shows that the SVC model gives quite high

posterior standard deviations in a small area in western Norway, south of *Sognefjorden*. This can be explained by that this both is an area where we have few observations (see Figure 1b) and where the original HBV map performs poorly and overestimates the true runoff.





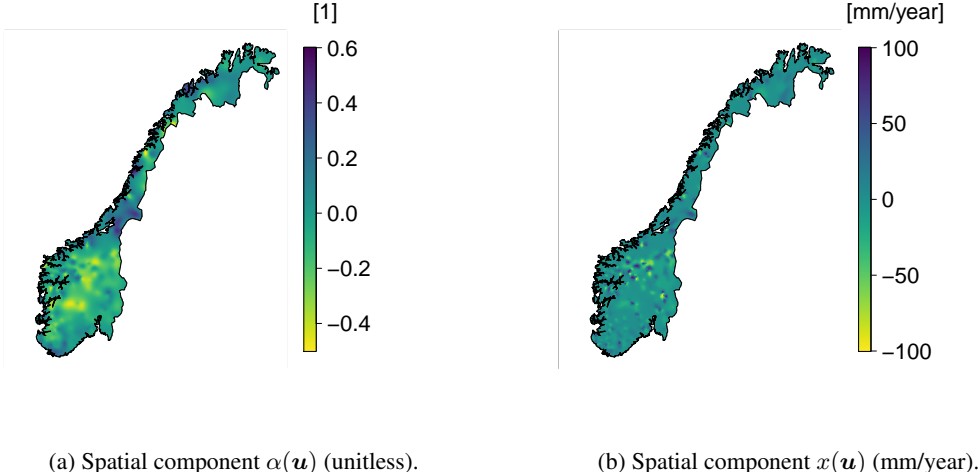

(a) Spatial component $\alpha(\boldsymbol{u})$ (unitless).      (b) Spatial component $x(\boldsymbol{u})$ (mm/year).

**Figure 6.** Posterior means for the two GRFs $x(\boldsymbol{u})$ and $\alpha(\boldsymbol{u})$ for SVC$_{\text{PP}}$.

In Figure 7 we present a scatter plot that shows the fit between the runoff map in Figure 5a and the observed mean annual runoff. The scatter plot is obtained by aggregating the grid nodes in Figure 5a to the catchment areas in Figure 1b. The results

show that the SVC$_{\text{PP}}$ map corresponds considerably better with the observed runoff for the fully gauged catchments than the original HBV map from Figure 2c. The original HBV map gave a correlation of 0.933 between the predictions and the observations for the fully gauged catchments, while the corrected SVC$_{\text{PP}}$ map gives a correlation approximately equal to 1.

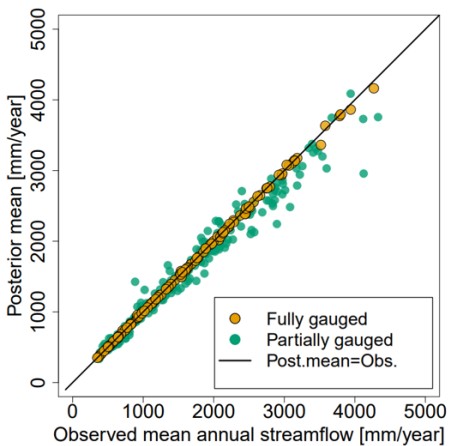

**Figure 7.** Scatter plot showing the predicted mean annual runoff for SVC$_{\text{PP}}$ and the observed mean annual runoff from fully gauged (orange) and partially gauged catchments (green). The predicted value is given by the posterior mean for $Q(\mathcal{A})$. Bear in mind that the mean annual runoff observations for the partially gauged catchments are based on 1-29 years of data from the 30 year time period of interest and must therefore be considered as approximations of the true mean annual runoff.





**Table 2.** Predictive performance for the cross-validation experiments when the target catchments are treated as ungauged (UG) and partially gauged (PG) for the HBV model, the suggested SVC model and for Top-Kriging (TK). Recall that subscript PP refer to the geostatistical preprocessing step, i.e. preprocessed data from partially gauged catchments are used in both Top-Kriging and the SVC approach. The results from the geostatistical preprocessing method (PP) are also included as a reference (without any further analysis) for a better understanding of the other results. The best method for each evaluation criterion is marked in bold.

| | | Ungauged target (UG) | | | Partially gauged target (PG) | | |
|---|---|---|---|---|---|---|---|
| | HBV | SVC$_{PP}$ | TK$_{PP}$ | PP | SVC$_{PP}$ | TK$_{PP}$ | PP |
| RMSE (mm/yr) | 394 | **315** | 350 | 389 | 166 | 181 | **134** |
| ANE | 0.180 | **0.111** | 0.125 | 0.192 | 0.054 | 0.053 | **0.047** |
| NSE | 0.815 | **0.881** | 0.854 | 0.771 | 0.968 | 0.961 | **0.978** |
| CRPS (mm/yr) | 235 | **145** | 173 | 209 | 73 | 77 | **71** |
| Coverage (90 %) | × | 0.83 | **0.91** | 0.96 | 0.95 | **0.94** | 1 |

We also investigated the correlation between the map and the observed runoff for the partially gauged catchments where we only have 1-29 years of measurements in the 30 year time period of interest (Figure 7). For these catchments, the original HBV
model gave a correlation of 0.917. The SVC$_{PP}$ map gives correlation 0.986. The correlations and Figure 7 indicate that the SVC$_{PP}$ map provides a better fit for the partially gauged catchments than the original HBV map. Here, we can not be entirely sure because the underlying observations from the partially gauged catchments in Figure 7 only are approximations of the true runoff between 1981-2010, computed based on 1-29 annual observations from this time period. It is however a good sign that the fit for the partially gauged catchments (green) is not as good as for the fully gauged catchments (orange). Since we don't
know the underlying truth for the partially gauged catchments, the SVC model should not necessarily reproduce the observed value.

## 6.2    Cross-validation for ungauged and partially gauged catchments

In Table 2 we present the results from the cross-validation assessment described in Section 5.1.2. Here, we compare our geostatistical model to the process-based HBV model and to the purely geostatistical Top-Kriging (TK) method in terms of
predicting mean annual runoff for ungauged (UG) and partially gauged (PG) catchments for 1981-2010. For reference, we have also included an evaluation of the prediction results provided by the preprocessing method alone (PP) without doing any further analysis. The PP results come from the purely geostatistical method from Roksvåg et al. (2020).

For ungauged catchments (UG), we find that the RMSE of our SVC$_{PP}$ method is 20 % lower than the RMSE of the HBV model. Compared to Top-Kriging, the SVC$_{PP}$ model gives 10 % lower RMSE. The ranking between the models is the same also
for the ANE, NSE and CRPS. When it comes to uncertainty quantification, Top-Kriging gives the best uncertainty representation for ungauged catchments according to the 90 % coverage, with 0.91 coverage. However, SVC$_{PP}$ also performs acceptable with 0.83 coverage on a cross-validation performed on (only) 127 catchments.





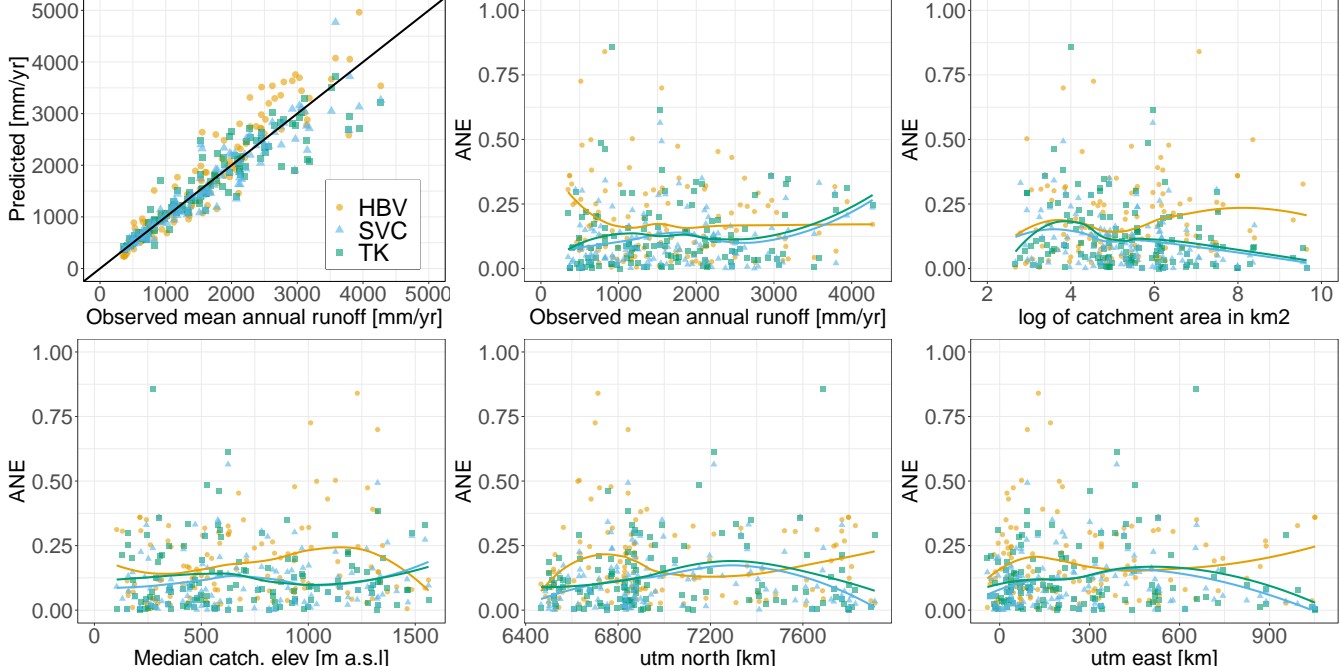

**Figure 8.** Predictive performance of the methods (HBV, SVC_PP and TK_PP) for predictions in ungauged catchments (UG) performed by cross-validation. The first plot shows the fit between the predictions and the observations for the methods. The remaining plots show the ANE for each of the 127 cross-validation catchments plotted against some selected catchment attributes; more specifically the observed runoff, catchment area, median catchment elevation, utm33 north and utm33 east. The fitted curves are regression splines (made by `geom_smooth()` in R) that make it easier to see trends in the predictive performance.

In Table A1 in the Appendix, we include the methods' predictive performance for ungauged catchments when not using the preprocessing step (SVC and TK). Hence, we only used observations from the 127 fully gauged catchments in the observa-
tion likelihood. These results give the same ranking between the methods as before, but with one exception: SVC performs approximately as good as Top-Kriging in terms of $90\%$ coverage, with coverages of 0.87 and 0.94 respectively. From Table A1 we also mark that the difference in performance between the SVC model and Top-Kriging is larger for this setting, where we omitted the short records, than for the setting where we included the short records. This is reasonable as we can expect the SVC model to be more robust than a purely data-driven model if the data availability is poorer. This was also a main motivation
for incorporating process-based simulations into a geostatistical model.

Further, we compared the predictive performance for ungauged catchments (UG) for the SVC_PP approach, the HBV model and Top-Kriging (TK_PP) across the study area and across catchment attributes in terms of the absolute normalized error (ANE). This is visualized for some selected catchment attributes in Figure 8. We see that the HBV model in general tends to over-estimate the mean annual runoff. It gives the highest ANE values in the south-western part of the country, and particularly
for catchments at higher elevations (800-1400 m a.s.l). The latter might be due to the interpolated precipitation product used





as input in the HBV-model, where orographic enhancement of precipitation is accounted for by an elevation gradient. Since precipitation gauging stations seldom are located at high elevations (Lussana et al., 2018), the precipitation is actually extrapolated to the highest altitudes giving rise to biases in the precipitation field. Figure 8 further shows that the two geostatistical approaches ($SVC_{PP}$ and $TK_{PP}$) perform better than the HBV model for catchments with mean elevations in the range 800-1400 m a.s.l.. This demonstrates that the SVC approach is able to compensate for its poor HBV input in these areas.

The lines in Figure 8 next show that Top-Kriging and $SVC_{PP}$ in general tend to follow the same trends across catchments attributes: For example do they both perform particularly well for catchments with large drainage areas, supporting existing results from Viglione et al. (2013) regarding the predictive performance of Top-Kriging. For catchments with large drainage areas, there are typically data from overlapping subcatchments available which makes areal referenced geostatistical models particularly appropriate. The two geostatistical approaches also perform well for catchments located in the eastern parts of Norway. In the south-eastern Norway we find catchments with larger drainage areas and most of them are located at relatively low elevations. The runoff in such catchments are typically easier to predict. The data availability is also good in the south-eastern parts of Norway, making geostatistical approaches particularly suitable. A trend describing differences in the predictive performance between Top-Kriging and the $SVC_{PP}$ approach is harder to see from Figure 8, but we notice that Top-Kriging (and the HBV model) in general produce more extreme ANE values than the SVC model.

So far in this subsection, we have only discussed the methods' ability to predict runoff in ungauged catchments. We now consider the results for the predictions in partially gauged (PG) catchments. Recall that for the PG case, there are 3 annual observations available from the target catchments (out of 30) and that these are preprocessed as described in Section 4.4 before further analysis in the mean annual runoff model. The results for the partially gauged catchments are shown in Table 2 and we see that we obtain a large reduction in the predictive performance for the $SVC_{PP,PG}$ case compared to the case when we have no data from the target catchments ($SVC_{PP,UG}$): The reduction in RMSE is 47 % when comparing these two settings. The improvement for $SVC_{PP,PG}$ compared to the HBV model is $58\%$. Compared to Top-Kriging, the $SVC_{PP}$ approach is slightly better in terms of RMSE, but approximately equally good in terms of ANE, NSE, CRPS and 90 % coverage. Table 2 also shows that the Top-Kriging estimates are substantially improved when including preprocessed short records from the target catchments in the likelihood (PG compared to UG for $TK_{PP}$).

The improved performance of $TK_{PP}$ and $SVC_{PP}$ for the PG case is mainly caused by the preprocessing procedure's ability to perform (very) accurate predictions of Norwegian mean annual runoff when a few annual observations are available. This can be understood from the results in Table 2 where we see that the input data provided by the preprocessing step (PP) alone gives predictions that are better than the predictions of the $SVC_{PP}$ and $TK_{PP}$ approaches. The improved results for $TK_{PP}$ and $SVC_{PP}$ however, show that the two geostatistical methods are able to incorporate the good performance of the PP method in their spatial interpolation procedures and that the SVC approach indeed can be used to combine both process-based data and data from fully gauged and partially gauged catchments.



# 7 Discussion

We have presented a geostatistical model for annual runoff that incorporates simulations from a process-based model through
a spatially varying coefficient and shown how short records can be included in the modeling by using the methodology from
Roksvåg et al. (2020) as a preprocessing step for partially gauged catchments.

In a preliminary study we tested models with only one spatial field, i.e. only $x(\boldsymbol{u})$ or $\alpha(\boldsymbol{u})$ was included in Equation (4).
These models performed quite well in terms of both posterior mean and posterior uncertainty for the Norwegian dataset, which
indicates that it for many study areas might be satisfactory to use a model with only one spatial field (i.e. similar to only
performing ratio interpolation or only residual interpolation). However, our preliminary experiments also showed that a model
with two spatial fields ($\alpha(\boldsymbol{u})$ and $x(\boldsymbol{y})$) often gave a more more realistic spatial distribution of uncertainty than a model with
only $x(\boldsymbol{u})$ or only $\alpha(\boldsymbol{u})$. Further, in Figure 6 we saw that the model was able to capture both short and long ranged processes
through its two fields, which can be a useful model property that can avoid that the model smooths out the process-based
covariate too much. In general, the importance of $x(\boldsymbol{u})$ compared to $\alpha(\boldsymbol{u})$ depends on the study area, the data availability and
the quality of the process-based input model.

Table 2 showed that Top-Kriging and the SVC approach both were able to exploit the preprocessing method's ability to
perform accurate predictions for partially gauged catchments. However, for these catchments TK$_{PP}$ and SVC$_{PP}$ performed
slightly poorer than the preprocessing input model alone (PG in Table 2). This is not necessarily a problem: The preprocessing
method (PP) is designed to be particularly suitable for record augmentation, while TK and SVC have other strengths. We also
did not want the SVC approach and Top-Kriging to put too much weight on the more uncertain preprocessed short records. The
latter was included in the model by specifying a larger (prior) observation uncertainty for the partially gauged catchments (0-23
% of the observed value) compared to the fully gauged catchments (0-6 % of the observed value). We have not tested how this
uncertainty specification affects the results, but in future work, the SVC$_{PP}$ model and TK$_{PP}$ might be improved by selecting
the observation uncertainty for the preprocessed data more carefully. The observation uncertainty for the partially gauged
catchments can e.g. be set independently of the fully gauged catchments and based on the record length of the short records.
An option could also be to use the predictive uncertainty of the preprocessing method to specify the (prior) measurement
uncertainty for the partially gauged catchments in the SVC model and Top-Kriging.

In this paper we presented a framework for *mean annual runoff* which is one of several key flow indices. The SVC framework
can be used for other flow indices as well, but the computational complexity makes it most suitable for flow indices of longer
temporal scale or for modeling long-term averages. The user should also know that the soft constraints imposed by Equation
(6) and the observation likelihood in Equation (7) assume a linear aggregation of runoff over the grid nodes that define the
catchment discretization. This is reasonable for mean annual runoff, but not for all hydrological variables. If the modeler wants
to avoid this model property, two simple modifications of the model are possible:

**m1:** Make the runoff observations point referenced instead of areal referenced by using e.g. the catchment centroids as the
target locations. This means omitting the integral in Equation (6) and letting $Q(\mathcal{A}) = q(\boldsymbol{u}_{\mathcal{A}})$ where $\boldsymbol{u}_{\mathcal{A}}$ is the centroid of
catchment $\mathcal{A}$ and $q(\cdot)$ is point runoff as defined in Equation (4). The drawback of this alternative is that the model will weight





observations from subcatchments similarly as observations from non-overlapping catchments and provide a poorer uncertainty representation.

**m2**: Adding more flexibility to the model by adding more covariates or random noise outside the integral in Equation (6). This
alternative preserves the areal representation of catchments, but makes it easier to violate the water balance constraints over nested subcatchments.

A potential weakness of the model proposed in this article, is that it uses a Gaussian likelihood. Hence, the model can provide negative runoff predictions. This can happen particularly if the flow index and the corresponding study area have many runoff
observations close to zero. The possibility of negative runoff is another argument for using the SVC model mainly for flow indices of a longer temporal scale or for modeling long-term averages.

To avoid negative runoff predictions there are some modifications of the model that can be done: For example is it possible to log transform the runoff data before performing the analysis, but this requires that we model the runoff observations as point referenced as proposed in **m1**. The reason is that the sum in Equation (6), which is related to how we model catchment
runoff, does not makes sense for log transformed runoff data. Other sources for negative predictions are the discretization of the study area and the mesh used for making inference (see Section 4.6). The discretization and the mesh should be dense enough to capture the spatial variability in the study region. In this article the HBV simulations and the associated catchment discretization were delivered on a $1 \times 1$ km grid and no negative values were produced.

In the proposed model, we used the model from Roksvåg et al. (2020) as a preprocessing step to exploit short records. The
preprocessing step can only be expected to improve the predictions for the partially gauged catchments if the study area and the flow index of interest are driven by runoff patterns that are repeated over time, like the mean annual runoff in Norway. If this is not the case, the preprocessing step performs a more classical form of spatial interpolation and can be omitted to save computational time. The performance of the preprocessing step over different study areas and target variables is further discussed in Roksvåg et al. (2020).
Figure 7 showed that the SVC$_{PP}$ gave a very good fit for the 127 fully gauged catchments, almost entirely reproducing the actual observed mean annual runoff in the resulting gridded map. We emphasize that the proposed method is not guaranteed to reproduce the observed value with the precision we saw in this case study. How good the fit becomes for the fully gauged catchments depends on e.g. the data quality, the gauging density and the complexity of the spatial variability of the underlying hydrological processes.
In Norway the gauging density is moderate. We expect the suggested SVC model to outperform purely geostatistical methods like Top-Kriging for gauging densities that are low to moderate. For data sparse areas, the process-based information provided by the HBV model is probably more important than in data dense areas. This claim is based on intuition about the models under discussion, but is also indicated by our results: Top-Kriging is closer to the SVC model in predictive performance for the dataset where we use data from 411 catchments (UG in Table 2) than for the reduced dataset where we only use data from
127 catchments (Table A1 in the Appendix ). This further suggests that if the gauging density is large relative to the spatial variability, a purely geostatsitical approach will perform as good as the SVC model.





Whether the suggested framework performs better than a purely geostatistical method is of course also connected to the quality of the process-based input model and the calibration procedures performed on the hydrological product. However, we have certainly shown that is possible to improve a process-based hydrological product by using the suggested framework: All
experiments showed that the SVC approach improved the predictions compared to the original HBV simulations. This means that the SVC model can be considered as an objective approach for correcting the simulations from a process-based model, and consequently reduce the need for more subjective, manual corrections.

## 8 Conclusions

In this article we have presented a Bayesian geostatistical model for annual runoff estimation that incorporates simulations from
a process-based hydrological through a covariate whose regression coefficient is allowed to vary in the study area according to a Gaussian random field. A preprocessing step for including short records in the modeling was also suggested such that the model could exploit data from both fully gauged and partially gauged catchments.

The model was evaluated by predicting mean annual runoff data for Norway (1981-2010), and simulations from the process-based HBV model were used to make the covariate. The results showed that the suggested framework outperforms a purely
process-based model when predicting runoff in ungauged and partially gauged catchments. The reduction in RMSE was 20 % for ungauged catchments and 58 % for partially gauged catchments. The increased predictive performance obtained compared to a purely process-based model is connected to the quality of the process-based product and the calibration procedures performed on it. However, all results show that the suggested framework is able to improve the predictions from a process-based model. This means that the approach can be used as a objective method for correcting process-based runoff maps relative to
data, which can reduce the need for more subjective, manual corrections. The large reduction in RMSE for partially gauged catchments also demonstrates that the preprocessing method from Roksvåg et al. (2020) can be incorporated into the proposed model to exploit short records.

Furthermore, the suggested model gave a 10 % lower RMSE than a purely geostatistical method (Top-Kriging) when predicting runoff in ungauged catchments. Particularly if the gauging density is low to moderate, we expect the suggested framework
to outperform purely geostatistical models. For partially gauged catchments that had a few annual streamflow observations available, a purely geostatistical method performed equally well (Top-Kriging) or slightly better (PP) than the proposed approach. It is not surprising that a purely data-driven framework performs well in areas where there actually are data. However, since most study areas consist of a mix of ungauged, fully gauged and partially gauged catchments, the proposed SVC model stands out as a good approach for making a consistent gridded runoff map for a larger area.

*Author contributions.* TR: Wrote R code, did the experiments, wrote the majority of the paper and made figures. IS: Came up with initial ideas, contributed with discussion throughout the work and with ideas for experimental set-up. KE: Provided the data, contributed with





discussion throughout the work, particularly regarding the hydrological context and the data quality, and contributed to the writing of Chapter 2.

*Competing interests.* No competing interests are present.

*Code and data availability.* The data and code used in this study can be provided by the main author upon request.

*Acknowledgements.* We would like to thank Stein Beldring at NVE for providing the gridded HBV product and for valuable discussions around the work. The project is funded by the Reasearch council of Norway, grant number 250362.





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

## Appendix A: Results when omitting short records

We repeat the experiments from Section 5.1.1 and 5.1.2 for ungauged catchments, but we only use observations from the 127
fully gauged catchments in Figure 1a. The runoff data from the partially gauged catchments are simply removed from the
analysis and the workflow is as in Figure 4 without performing the optional preprocessing step. The experiments are included
to show that the SVC model works regardless of preprocessing.

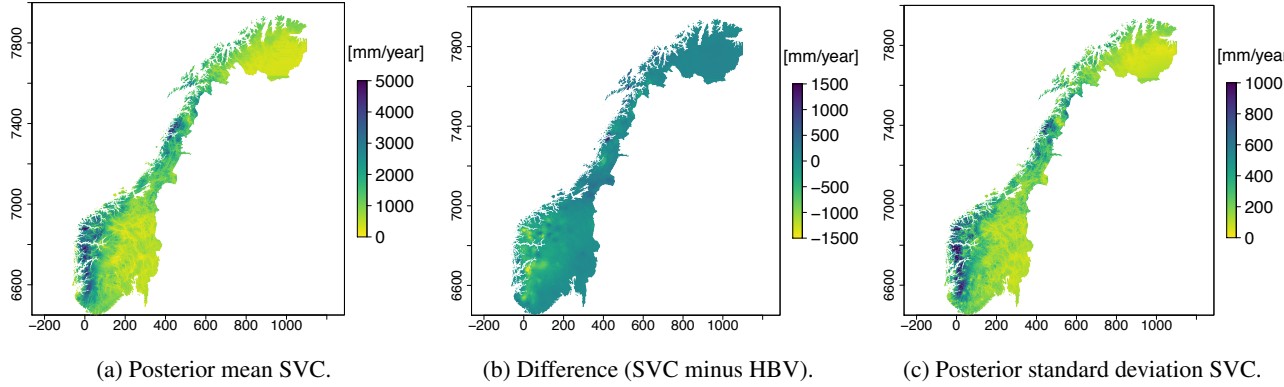

(a) Posterior mean SVC.     (b) Difference (SVC minus HBV).     (c) Posterior standard deviation SVC.

**Figure A1.** Posterior mean of $q(\boldsymbol{u})$ for all grid nodes, difference between the new map and the original HBV map and posterior standard deviation of $q(\boldsymbol{u})$. The model is fitted without including short records in the observation likelihood, i.e. without using the preprocessing step.

The runoff map provided by the SVC model when not using short records, as described in Section 5.1.1, is shown in Figure
A1. The maps look similar to the maps in Figure 5, but the posterior uncertainty is larger in western Norway in Figure A1c.
The reasons are that there are less observations available from western Norway in the dataset consisting only of fully gauged
catchments and that this is an area with large deviance between the original HBV map and the observed streamflow.

In Figure A2 we show the fit between the observed runoff and the runoff predicted by the map in Figure A1a. The fit is
very good for the fully gauged catchments, as before. The fit is also improved for the partially gauged catchments compared
to the original HBV map in Figure 2a. Here, the original HBV model gave correlation 0.917 between observed and predicted
values, while the map in Figure A1a gives correlation 0.924. However, when short records and preprocessing were included in





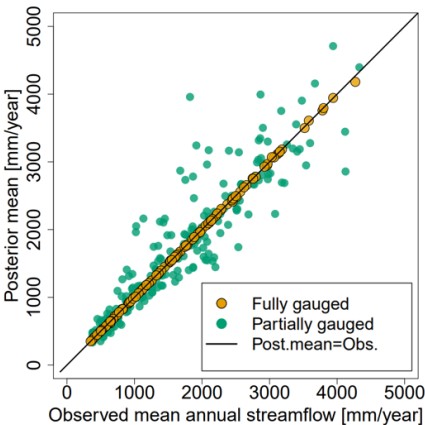

**Figure A2.** Scatter plot showing the predicted mean annual runoff (posterior mean of $Q(\mathcal{A})$) for SVC and the observed streamflow from fully gauged and partially gauged catchments when short records are omitted from the likelihood.

**Table A1.** Predictive performance for cross-validation when the target catchments are treated as ungauged (UG) for the HBV model, the suggested SVC model and for Top-Kriging (TK). Short records are omitted from the observation likelihood and the preprocessing step is not performed. The best method for each evaluation criterion is marked in bold.

|  | UG | | |
| --- | --- | --- | --- |
|  | HBV | SVC | TK |
| RMSE (mm/yr) | 394 | **320** | 381 |
| ANE | 0.180 | **0.135** | 0.176 |
| NSE | 0.815 | **0.878** | 0.827 |
| CRPS (mm/yr) | 235 | **156** | 211 |
| Coverage (90 %) | × | **0.87** | 0.94 |

the analysis, the correlation was 0.986 (SVC$_{PP}$ in Figure 7). This illustrates the reduced predictive performance when omitting short records from the analysis in Norway and in countries with similar temporal trends in annual runoff.

Finally, we present the cross-validation results when using the dataset that only consists of fully gauged catchments, as described in Section 5.1.2. The results are summarized in Table A1. Again the SVC model performs considerably better than the HBV model and Top-Kriging in terms of RMSE, ANE, NSE and CRPS. Mark that the difference in performance from Top-Kriging is larger for this dataset (for UG), compared to when using the larger dataset that included short records (Table 2). This is reasonable as we can expect purely data-driven methods to increase their performance when more data are available.