# Peer review of "A geostatistical spatially varying coefficient model for mean annual runoff that incorporates process-based simulations and short records"

_Hydrology and Earth System Sciences, 2021_

## Author Comment (AC2)

**Response to Anonymous referee 2**

We would like to thank Anonymous referee #2 for his/her insightful review and for bringing up several interesting topics for discussion. Most of the comments are related to the model components and/or about motivating model assumptions. We will clarify this part in our revised MS. In addition we will edit/rewrite the awkward sentences that are mentioned in the review.

See our detailed answers to referee #2 below.

Kind regards,
Thea Roksvåg and co-authors.
* * *
**R2**: *"A potential weakness of the method, which has been mentioned by the authors, is that the model does not prevent negative run-off predictions in some (unlikely situations). This is due to the Gaussian likelihood and Gaussian GRF. The authors mention that log-runoff could be used instead, but then linearity of Eq. (6) is lost, which is an impediment. Another way of preventing negative predictions would be to use log-Gaussian likelihood and log-Gaussian random fielsd for x(u) and α(u) in (4). This would be a marginal change, since INLA/SPDE allows for log-gaussian likelihood and LG random fields at almost no cost. As a result, predictions for x and α would always be positive. I wonder how this would work. Ideally, I'd like the authors to try this option, but I'd be happy if they only discuss this possibility."*

**Response:**
It should be possible to make a model like this in inla, to avoid negative predictions. The drawback is that a log GRF will make it more difficult to interpret the two spatial fields. As the paper already is quite long, we will add this to the discussion, as possible further work.

**R2:** *"The GRFs x(u) and α(u) are independent. This assumption is never clearly stated and it is not discussed. Is this a reasonable assumption? Is this an assumption you could check or validate? How useful/difficult would it be to relax this assumption?"*

**Response:**
We will clearly state that *x(u) and α(u)* are assumed to be independent, and discuss whether this is reasonable. We can compare *x(u) and α(u) to investigate whether they are independent.*

To relax the independence assumption we could include a third spatial field that is both included in the factor multiplied with h(x) and added to the model (multiplied with a scaling coefficient). This will result in a model that is harder to interpret, but might give better results. A drawback of this option, is that an additional spatial field will significantly increase the computational complexity of the model, and the computational complexity is already quite high. The increased computing time is

probably not worth it, as the current model gives quite good results as it is.

**R2:** "The GRFs x(u) and α(u) are assumed to be stationary. Are you able to check that this assumption is supported by the data?"

**Response:** The GRFs x(u) and/or α(u) might be non-stationary. The spatial dependency structure of precipitation and runoff can change with for example elevation. We would e.g. expect the spatial range of x(u) to decrease with elevation. Other non-stationary effects could also exist.

We could investigate the non-stationarity of the spatial fields by fitting non-stationary models. From this we can see whether the non-stationary effects are significant. We could also compute empirical ranges and variances from the data, for different areas. However, modelling x(u) and/or α(u) as stationary is a choice we have made based for the following reasons:
* Modelling x(u) and/or α(u) as non-stationary, would introduce additional parameters to the model which will represent an increase in computational complexity.
* The type of non-stationarity can be difficult to identify from the data.
* According to Fuglstad et al (2015), non-stationary processes can in many cases be modelled by stationary models.
* Even if the underlying process is non-stationary, we think that our model should work well because we have two spatial fields. Together, x(u) and α(u) give a flexible model that is able to capture different dependency structures in the data.

**R2:** *"To my knowledge, the product of an exponential variogram with a fractal variogram is not a valid variogram. However, the product of an exponential covariance function with a fractal variogram might be a valid variogram. Please double-check and provide references if necessary. "*

**Response:** We fitted the default variogram type in the rtop package. According to the package documentation (https://cran.r-project.org/web/packages/rtop/rtop.pdf) this is a *"multiplication of a modified exponential and fractal model"* (model="Ex1").

**R2:** *"Regarding the results: is it really desirable to get a correlation of 1 between measures and predictions? I would relate this to the fact that the coverage is 83%, which shows that the SVC is over-confident in the UG setting. Please comment."*

**Response:**
It is not necessarily desirable to get a correlation of 1 (as for the orange points in Fig 7), and the coverage of 83 % indeed shows that the SVC is slightly over-confident in the UG setting. By using a model with two spatial fields, we get a model that is quite flexible. This can explain why we sometimes get correlations close to 1. Furthermore, the HBV model aims at getting perfect predictions, which would imply a correlation of 1.

Whether a correlation of 1 is desirable or not, depends on what we want from our runoff map. If it is important that the map is correct in areas where we have data, a correlation of 1 between the map and the data is good. Alternatively, we could accept a lower correlation, and get a model that might perform better in ungauged areas (more spatial smoothing).

After the MS was prepared, we did some more experiments with the SVC model, where we used a larger dataset with 180 gauged catchments and 450 partially gauged catchments. We also used a different hydrological model (WASMOD instead of the HBV model). In these experiments, the correlation between the observations and predictions were lower than 1. Hence, whether we get a correlation of 1 depends on the dataset. In general, we would expect a lower correlation if we have more data as it then becomes more difficult for the model to fulfil all data "constraints". In addition, the correlation depends on the hydrological product used and how it is calibrated.

In the discussion, on line 720, we briefly mention that we cannot expect a perfect fit (correlation 1) between the observations and the runoff map. In a revised version of the manuscript, we can rewrite/extend this part, and/or rewrite the results section (around line 595 and 600).

**R2:** Comments about awkward sentences and spelling mistakes.
**Response:** We will rewrite/edit these sentences.

**References:**
Geir-Arne Fuglstad, Daniel Simpson, Finn Lindgren, Håvard Rue,
Does non-stationary spatial data always require non-stationary random fields?,
Spatial Statistics,
Volume 14, Part C, 2015,
Pages 505-531,
ISSN 2211-6753,
https://doi.org/10.1016/j.spasta.2015.10.001.
(https://www.sciencedirect.com/science/article/pii/S2211675315000780 )

---

## Author Response (AR1)

Dear Editor,

We would like to thank the two referees for their constructive reviews. We were pleased by the level of detail they put in their comments. This made the revision process easier and the manuscript better.

The major concern of referee 1 was that the MS was too long and that some parts were repeated several times. The referee had several suggestions on which parts we could remove to improve the MS. We have followed most of the suggestions and also removed other parts of the article to reduce its length. The length of the original MS was 29 full pages, when not including references and the Appendix. The length of the revised paper is approximately 25 pages (without references and the Appendix).

Another concern of referee 1 was that some critical information was only mentioned in the last part of the article and not in the first. One example is that we only discussed that the model can give negative estimated runoff in the last part of the article. This is now discussed earlier, in the "methods" part. In the revised MS we also mention that the model is particularly suitable for mass conservative variables in the "methods" section. This was originally only mentioned in the "discussion" part.

Referee 2 had several interesting questions about our model assumptions. We have added some discussion about model assumptions in the revised MS and clarified our model assumptions further.

Again we would thank the two referees for their valuable contribution. It has improved the quality of the paper.

See detailed answers to the referee comments below.

Kind regards,
Thea Roksvåg and co-authors

**Detailed responses to the referee comments**

**Referee 1**

**R1:** "Row 17 and following. The Introduction is too long and the description of the work performed in this study is split into several parts. I suggest removing some sentence (as rows 21-22, "The temporal variability of runoff can also be used to study runoff's sensitivity to climate change") that do not provide additional information related to this work. I also suggest to summarize the work and the main novelties at the end of the

paragraph."

**Response:** We have removed this sentence and several other sentences. The introduction is still split into several parts, but many of the parts are now shorter, so it should be easier to follow. The main objectives are summarized in the end of the introduction.

**R1:** "Row 121. You used a ratio equal to 0.2. Why? Please provide a reference or a motivation."
For catchments with a reservoir capacity smaller than 0.2, the annual changes in water storage is small compared to the annual inflow volume. This can be explained by a strong seasonality in the reservoir storage, i.e. the annual changes in reservoir storage using the hydrological years is much smaller than the reservoir capacity. Checking for a subset of these catchments, we found that the standard deviation of annual changes in reservoir storage was less than 2% of annual inflow.

We have included this motivation on line 99-102.

**R1:** Row 133. Please mention the spatial interpolation that you used.
**Response**: We have used the spatial interpolation method described in subsection 3.3.2. However, we have removed the sentence in question, since the preprocessing step is described later, in subsection 3.3.2 and subsection 4.4.

**R1:** "Row 226. Please explain what SPDE is."
**Response:** We have added what SPDE stands for and changed the order of the sentences such that it becomes clearer that we use the SPDE approach for computational reasons. We also refer to Section 4.6 where we describe what the SPDE approach is. See line 175.

**R1:** Row 236. "x" is an estimated variable: please insert the "hat".
**Response:** We don't use a hat here, because we refer to the underlying field we are trying to estimate, and not the estimator itself.

**R1:** Rows 279 – 284. I suggest to remove them.
**Response:** We have made this part shorter, but we have kept some of it to make it clear to the reader that we in the next subsections are building up a three stage hierarchical model according to the methodology in Section 3.1.

**R1:** Row 329. Is the requirement of having positive runoff satisfied? If not, please provide additional information about how you managed this issue. If I am not wrong, you provide additional information about this only in row 713. I suggest to anticipate this statement.
**Response:** We have added some discussion around negative runoff in Section 4.2, lines 269-277. We have also kept some of the discussion regarding negative runoff in Section 7, since this is an important model weakness.

In addition, we have added to two sentences on lines 252-254 about how the areal formulation is suitable for mass conservative variables. This was originally a part of the discussion, but because it is an important model property, we mention it earlier.

**R1:** Row 372. "Credibility interval" or "confidence interval"?

Row 409. I think you were referring to "confidence interval" and not to "credible interval".
**Response:** Credible intervals are the Bayesian version of confidence intervals: https://en.wikipedia.org/wiki/Credible_interval

**R1:** Rows 464 – 467. Please remove them or move them before. It is not advisable to mention the goals of the study after 18 pages of text. You already provided some information about the goals in several parts of Section 1. I suggest to merge everything, providing a more detailed and structured statement.
**Response:** The goals mentioned here are removed, and written more clearly in the last part of the introduction. See lines 74-83.

**R1:**
"Rows 14-15. I suggest removing from the abstract the sentence "It is not surprising that purely geostatistical methods perform well in areas where we have data". It is already expected and does not provide additional information to the reader. Moreover, it makes the article less robust."
"Row 35. Top-Kriging is used to model only discharge and not other referenced data, as mentioned. Please correct."
"Row 55. The correct and most used name is "kriging with external drift" and not "external drift kriging"."
Row 58. "Was estimated with a …".
Row 59. Please explain what "g(.)" is.
Row 81. Please explain here what GRF is, considering that it is the first time that you mention this acronym.
Row 99. Please explain what INLA and SPDE are.
Rows 124-125. Please check these rows, I am not sure that the grammar is correct.
Description of Figure 1. Please use "UTM" instead of "utm".
Rows 164 – 165. I suggest removing "However, most of the 141 calibration catchments probably coincide with the 127 fully gauged catchments in Figure 1a". If you are not sure about it, it is not advisable to mention it.
Rows 178-179. I suggest to remove them.
Row 206. Is the square bracket correct? Or is it a typo?
Row 209. I suggest to remove "will".
Row 224. Please insert the equation in a new line, by assigning number (3).
Rows 231-233. I suggest to remove them.
Row 235. I suggest to shorten the sentence in "Kriging is used to …".
Row 249. Please correct with "According to Viglione et al. (2013) and Blöschl et al. (2013) …".
Rows 307 – 310. You already mentioned this before. I suggest to remove this part.
Row 313. Why is "areas" in italics?
Row 340. Please rewrite, removing "bear in mind".
Row 352. Please remove or rewrite "because Norway is a diverse country when it comes to runoff generation". For me, it is not clear what you meant.
Equation 14. I suggest to insert it as three separate equations. Please change the blue font to black font.
Rows 470 – 471. Please remove "These are observations from 127 fully gauged catchments from 1981-2010 and 284 partially gauged catchments from 1965-2010". You already mention it.
Row 491 and following rows. The bold font is not necessary.
Rows 561 – 563. I suggest to remove them.
Figure 6. Please remove the "[1]" above the first colorbar. Please insert the axis (with coordinates).

Figure 7. I suggest to use "UTM" instead of "utm".
Row 637. Is "do" necessary?
Row 671. Please remove the second brackets after x(y).
Row 702. Please remove the empty row.
**Response:** We have revised the paper according to the above comments.

**Referee 2**

**R2:** A potential weakness of the method, which has been mentioned by the authors, is that the model does not prevent negative run-off predictions in some (unlikely situations). This is due to the Gaussian likelihood and Gaussian GRF. The authors mention that log-runoff could be used instead, but then linearity of Eq. (6) is lost, which is an impediment. Another way of preventing negative predictions would be to use log-Gaussian likelihood and log-Gaussian random fields for x(u) and α(u) in (4). This would be a marginal change, since INLA/SPDE allows for log-gaussian likelihood and LG random fields at almost no cost. As a result, predictions for x and α would always be positive. I wonder how this would work. Ideally, I'd like the authors to try this option, but I'd be happy if they only discuss this possibility.

**Response:** It should be possible to make a model like this in inla, to avoid negative predictions. The drawback is that a log GRF will make it more difficult to interpret the two spatial fields. We have not added any new experiments related to this in the revised paper, due to its length, but we have mentioned the possibility in Section 4.2, lines 272-275.

**R2:** The GRFs x(u) and α(u) are independent. This assumption is never clearly stated and it is not discussed. Is this a reasonable assumption? Is this an assumption you could check or validate? How useful/difficult would it be to relax this assumption?

**Response:** x(u) and alpha(u) are assumed *conditionally* independent a priori. In the revised MS, we have added a sentence where we emphasize this. See lines 236-237.

To relax the independence assumption, we could include a third spatial field that is both included in the factor multiplied with h(x) and added to the model (multiplied with a scaling coefficient). This will result in a model that is harder to interpret, but might give better results. A drawback of this option, is that an additional spatial field will significantly increase the computational complexity of the model, and the computational complexity is already quite high. The increased computing time is probably not worth it, as the current model gives quite good results as it is.

Also note that the *posterior* estimators for x(u) and alpha(u) probably are dependent, just like the intercept and the coefficients in linear regression.

**R2:** The GRFs x(u) and α(u) are assumed to be stationary. Are you able to check that this assumption is supported by the data?

**Response:** The GRFs x(u) and/or α(u) might be non-stationary (and an-isotropic). We could investigate the non-stationarity of the spatial fields by fitting non-stationary models, to see whether the effects are significant. We could also compute empirical ranges and variances from the data, for different areas. However, modelling x(u) and/or α(u) as stationary is a choice we have made for several reasons:

Modelling x(u) and/or α(u) as non-stationary, would introduce additional parameters. This would increase the computational complexity. Furthermore, the type of non-stationarity can be difficult to identify from the data. According to Fuglstad et al (2015), non-stationary processes can in many cases be modelled by stationary models.

We emphasize that even if x(u) and α(u) are stationary fields, the component α(u)h(u) allows for runoff that is spatially inhomogeneous in terms of both mean and variance given the process-based product h(u).

We have not added extra discussion regarding non-stationarity to the MS, due to the paper length, but we have emphasized in Section 4.1.1 that our spatial fields are stationary. See lines 227 and 237. In addition, we emphasize that the model can give runoff predictions that are spatially inhomogeneous, even if the spatial fields are stationary. See line 229-230, and line 500.

**R2:** To my knowledge, the product of an exponential variogram with a fractal variogram is not a valid variogram. However, the product of an exponential covariance function with a fractal variogram might be a valid variogram. Please double-check and provide references if necessary.

**Response:** We fitted the default variogram type in the rtop package. According to the package documentation (https://cran.r-project.org/web/packages/rtop/rtop.pdf) this is a *"multiplication of a modified exponential and fractal model"* (model="Ex1"). We have added a reference to the package documentation in the paper. See line 418.

**R2:** *Regarding the results: is it really desirable to get a correlation of 1 between measures and predictions? I would relate this to the fact that the coverage is 83%, which shows that the SVC is over-confident in the UG setting. Please comment.*

**Response:** It is not necessarily desirable to get a correlation of 1 (as for the orange points in Fig 7), and the coverage of 83 % indeed shows that the SVC is over-confident in the UG setting. By using a model with two spatial fields, we get a model that is quite flexible. This can explain why we sometimes get correlations close to 1. Furthermore, the HBV model aims at getting perfect predictions, which would imply a correlation of 1.

Whether a correlation of 1 is desirable or not, depends on what we want from our runoff map. If it is important that the map is correct in areas where we have data, a correlation of 1 between the map and the data is good. Alternatively, we could accept a lower correlation, and get a model that might perform better in ungauged areas (more spatial smoothing). We can affect the fit by changing the prior uncertainty of the model.

We have added some discussion around this topic on lines 602-605.

**R2:** Comments about awkward sentences and spelling mistakes.
**Response:** We have rewritten/edited/removed these sentences.

**References:**
Geir-Arne Fuglstad, Daniel Simpson, Finn Lindgren, Håvard Rue,

Does non-stationary spatial data always require non-stationary random fields?,
Spatial Statistics,
Volume 14, Part C, 2015,
Pages 505-531,
ISSN 2211-6753,
https://doi.org/10.1016/j.spasta.2015.10.001.
(https://www.sciencedirect.com/science/article/pii/S2211675315000780 )

---

## Author Response (AR2)

Dear Editor,

We have revised the manuscript according to the reviewer report. See the answers below for details. We would like to thank you and the two reviewers for your contribution. Your feedback has been very valuable in improving the quality of the paper.

Kind regards,
Thea Roksvåg and co-authors.

**Response to the reviewer comments (report #1):**

**Reviewer comment:**
162: What about the parameter $\nu$?
225-233: the parameter $\nu$ is never discussed. I guess it has been set when using SPDE/INLA, but the authors should indicate it more clearly

**Response:** nu is often not estimated in INLA applications, but set to a fixed value. We have added an explanation on page 8, line 180.

**Reviewer comment:**
220 and following: I wonder why the prior on $\beta_0$ and $\beta_1$ are shown here instead of Section 4.3

**Response:** In INLA, beta0 and beta1 are treated as a part of the latent field for technical reasons, but you could also think of them as parameters. We have moved the beta0 and beta1 prior specifications to Section 4.3 as suggested.

**Reviewer comment:**
237: In response to my comment on the independence of x(u) and $\alpha(u)$, the authors have added a sentence stating that they are conditionally independent. My question is: do you mean conditionally on h(u)? Recall however that h(u) is not a random quantity, but a (deterministic) covariate that contains the simulated value generated by the process-based hydrological model. Since h(u) is deterministic, the correct wording is that x(u) and $\alpha(u)$ are independent.

**Response:**
They are independent conditionally on their parameters, i.e. the sigma and the range. This is now clarified on page 10, line 238.

**Reviewer comment:**
257: I failed to see why h(A) should be computed; We do need the values h(u) for $u \in L\_A$, bu why do we need the sum h(A)?

**Response:** We have revised this sentence. See page 11, line 257.

**Reviewer comment:**
Section 4.1.1: notice that with this modelling, q(u) could also be negative (in theory; I guess it never happened in practice).
269: also true for q(u)

**Response:** We have added that also Equation (5) can contribute to negative runoff on page 11, line 270.

**Reviewer comment:**
333: ideally, $s_i^{PP}$ should depend on the length of the series

**Response:**
Yes, agreed. We have now mentioned this on page 13, line 340 (in addition to be discussed in the discussion section).

**Typo list:**
We have corrected the typos mentioned here.